# Detection of Outflow of Formaldehyde and Glyoxal from the African continent to the Atlantic Ocean with a MAX-DOAS Instrument

Lisa K. Behrens[1], Andreas Hilboll[1,2], Andreas Richter[1], Enno Peters[1,a], Leonardo M. A. Alvarado[1], Anna B. Kalisz Hedegaard[3,1], Folkard Wittrock[1], John P. Burrows[1], and Mihalis Vrekoussis[2,1,4]

[1]Institute of Environmental Physics (IUP-UB), University of Bremen, Bremen, Germany
[2]MARUM - Center for Marine Environmental Sciences, University of Bremen, Bremen, Germany
[3]DLR - Institute of Atmospheric Physics, German Aerospace Center, Oberpfaffenhofen-Wessling, Germany
[4]Energy, Environment and Water Research Center (EEWRC), The Cyprus Institute, Nicosia, Cyprus
[a]now at: DLR - Institute for protection of maritime infrastructures, German Aerospace Center, Bremerhaven, Germany

**Correspondence:** Lisa K. Behrens (lbehrens@iup.physik.uni-bremen.de)

**Abstract.** Trace gas maps retrieved from satellite measurements show enhanced levels of the atmospheric volatile organic compounds formaldehyde (HCHO) and glyoxal (CHOCHO) over the Atlantic Ocean. To validate the spatial distribution of this continental outflow, ship-based measurements were taken during the project **C**ontinental **O**utflow of **P**ollutants towards the **MA**rine t**R**oposphere (COPMAR). A Multi-AXis Differential Optical Absorption Spectrometer (MAX-DOAS) was operated on board the research vessel (RV) Maria S. Merian during the cruise MSM58/2. This cruise was conducted in October 2016 from Ponta Delgada (Azores) to Cape Town (South Africa), crossing between Cape Verde and the African continent. The instrument was continuously scanning the horizon looking towards the African continent. Enhanced levels of HCHO and CHOCHO were found in the area of expected outflow during this cruise. The observed spatial gradients of HCHO and CHOCHO along the cruise track agree with the spatial distributions from satellite measurements and MOZART-4 model simulations. The continental outflow from the African continent is observed in an elevated layer, higher than 1000 m, and probably originates from biogenic emissions or biomass burning according to FLEXPART emission sensitivities.

## 1 Introduction

### 1.1 Trace gases exported to ocean areas

Nitrogen oxides ($NO_x = NO + NO_2$), formaldehyde (HCHO), and glyoxal (CHOCHO) are important air pollutants. Enhanced levels of these species can be observed over anthropogenically and naturally polluted areas, e.g. industrialised areas or biomass burning regions in Africa (van der A et al., 2008, De Smedt et al., 2008, Lerot et al., 2010, Alvarado et al., 2014). However, all three trace gases have also been observed over remote ocean areas, for example, over the equatorial Atlantic Ocean close to the African continent (e.g., HCHO: Wittrock et al., 2006, De Smedt et al., 2008; CHOCHO: Vrekoussis et al., 2009, Alvarado et al., 2014, Lerot et al., 2010, Wittrock et al., 2006; $NO_2$: Richter and Burrows, 2002, Boersma et al., 2008, van der A et al., 2008). These enhanced values measured over the ocean are low, often close to the detection limits of satellite instruments (Vrekoussis et al., 2009). Consequently, shipboard measurements of HCHO and CHOCHO are needed to validate the retrieved

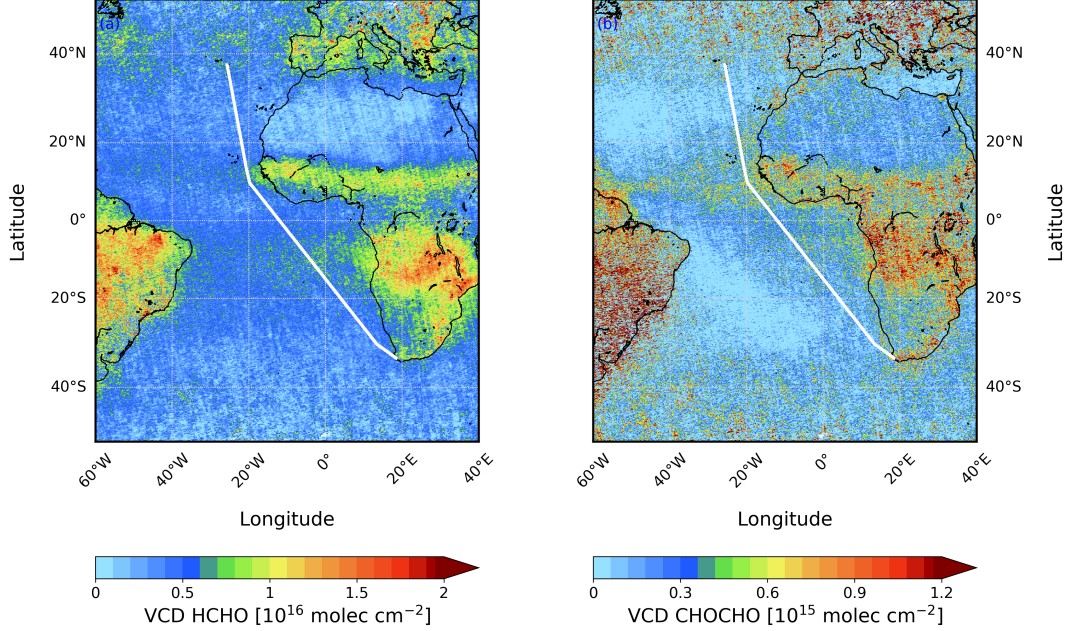

**Figure 1.** Monthly mean OMI satellite data for (a) HCHO and (b) CHOCHO for October 2016. In white the track of the cruise MSM58/2 is shown.

column amounts from nadir spaced based remote sensing instrumentation. Such validation is needed in support of the satellite measurements over remote ocean areas and together with model simulations, they can improve our understanding of horizontal and vertical distributions of these gases.

Tropospheric $NO_2$ is released to the atmosphere in large quantities from combustion, e.g., fossil fuels or biomass (Lee et al., 1997). The importance of the individual $NO_2$ sources varies by region (van der A et al., 2008). In Africa for example, biomass burning is known to be an important seasonally varying source but anthropogenic sources are also of significance.

HCHO is the simplest and most abundant aldehyde in the atmosphere. The dominant sources are biogenic compounds such as isoprene which can be oxidised and form HCHO (Koppmann, 2007). Primary emissions are from biomass burning and the
combustion of fossil fuels, but HCHO is also formed in the atmosphere from the oxidation of methane ($CH_4$) and non-methane hydrocarbons (Arlander et al., 1995). Due to the oxidation of $CH_4$, HCHO is not only found close to its source regions, but a global background concentration of HCHO exists with surface levels of $0.2 - 1.0$ parts per billion (ppb) in remote marine environments (Weller et al., 2000; Burkert et al., 2001; Singh et al., 2001). Furthermore, it is an important indicator of the photochemical activity for a region (De Smedt et al., 2008; Vrekoussis et al., 2010). Generally, HCHO is a short-lived species
($\sim$ 5 hours). Therefore, enhanced HCHO values are usually found close to the emission sources, e.g., industrial areas and tropical rainforests (De Smedt et al., 2008) and suppressed levels over the remote ocean are expected as observed in satellite observations (Fig. 1 a). However, Meyer-Arnek et al. (2005) showed that trace gases, including HCHO, can be transported into the area above the southern Atlantic Ocean originating from biomass burning and from biogenic emissions.

CHOCHO is the smallest alpha-dicarbonyl compound in the atmosphere. It is an intermediate species formed by the oxidation of volatile organic compounds (VOCs) or by direct emissions from biomass burning and the combustion of fossil fuels (Stavrakou et al., 2009). The most important precursor for CHOCHO is biogenic isoprene with a contribution of nearly 50 % globally (Fu et al., 2008). Similar to HCHO, CHOCHO is a short-lived species with a lifetime of about 3 h on global average (Myriokefalitakis et al., 2008) which is removed by photolysis, by reactions with OH radicals, and by dry and wet deposition (Fu et al., 2008). Consequently, it is expected that CHOCHO is observed close to the formation regions and "hotspot" areas of anthropogenic and biogenic emissions can be identified in global maps of CHOCHO columns derived from satellite measurements (Vrekoussis et al., 2010). Due to the short lifetime of CHOCHO, no transport over long distances is expected and therefore, no CHOCHO should be observed over remote ocean areas. However, several studies of satellite-derived CHOCHO columns reported enhanced CHOCHO levels over remote ocean regions (Fig. 1 b): Lerot et al. (2010) proposed the transport of continental CHOCHO precursors to remote oceanic regions, whereas Vrekoussis et al. (2009) suggested that the observed amounts of CHOCHO are related to upwelling regions and thus oceanic emissions. These regions have a large concentration of phytoplankton with high biogenic activity which could contribute to the emission of CHOCHO. Also Sinreich et al. (2010) concluded that the ocean must be a source of CHOCHO, because it was found up to 3000 km away from continental sources in the marine boundary layer. Stavrakou et al. (2009) included in their model a hypothetical additional biogenic source of CHOHCO over land. By including this additional source, they were able to improve the agreement between model and satellite observations over ocean which was related to an unknown CHOCHO precursor with a longer lifetime. Also Volkamer et al. (2015) considered an ocean source although their results indicate another source of CHOCHO.

## 1.2 Outflow of Aerosols

Outflow events from the African continent are regularly observed for aerosols, mainly between 0° and 20° N (Generoso et al., 2008). These aerosols originate from wind blown desert dust (Fairlie et al., 2007), but also from biomass burning emissions and secondary organic aerosol from biogenic emissions (Chung and Seinfeld, 2002). The transported aerosols affect the atmosphere (e.g., visibility, radiation budget, and air quality) and ecosystems world wide (Ridley et al., 2012). The dust emissions vary with time as they are influenced by different components of atmospheric circulation, e.g., Harmattan, Saharan heat low, and West African monsoon (Schepanski et al., 2017), and by surface conditions in the source regions, e.g., vegetation cover, and soil moisture (Ridley et al., 2012). If the dust aerosols are lifted, they can be transported westwards over long distances across the Atlantic. This transport is influenced by trade winds and shifts of the Intertropical Convergence Zone (ITCZ; Ridley et al., 2012; Schepanski et al., 2017). North of the Equator, the lowest aerosol optical depths (AOD) can be observed in northern hemispheric autumn (September, October, November: SON), when no biomass burning is present, and a peak in northern hemispheric summer (June, July, August; JJA; Ridley et al., 2012). Generally, the aerosol outflow is found in higher altitudes depending on the season; about 2000 m in winter (December, January, February; DJF) and about 5000 m in JJA (Ridley et al., 2012; Generoso et al., 2008).

Likewise, outflow events from Africa are regularly detected in the Southern Hemisphere. Anderson et al. (1996) observed outflow events of aerosols during a flight campaign in September/October 1992 where they found aerosols in the south Atlantic

Ocean originating from Africa as well as from South America. The aerosols from the African continent were detected in an altitude between 3000 and 4000 m.

## 1.3   Structure of this manuscript

This manuscript describes a study of HCHO and CHOCHO outflow from the African continent which was observed in MAX-DOAS measurements collected during a ship cruise in October 2016. The manuscript is structured as follows: In Sect. 2, the COPMAR project is described. In Sect. 3, the satellite and model data used for the analysis are introduced as well as the instrument used for the measurements. In Sect. 4, the measurement setup is evaluated using measurements of stratospheric $NO_2$. Furthermore, the results for the individual species are presented, discussed, and compared with model data such as FLEXPART simulations. Section 5 shows comparisons with previous studies. The manuscript ends with a summary and conclusions in Sect. 6.

## 2   The ship cruise – COPMAR project

The **C**ontinental **O**utflow of **P**ollutants towards the **MA**rine t**R**oposphere (COPMAR) project took place on board of the German research vessel (RV) Maria S. Merian as part of the cruise MSM58/2, which started in Ponta Delgada (Azores) on 08 October 2016, passed between Cape Verde and the African Continent, and ended on 25 October 2016 in Cape Town (South Africa; see Fig. 2; Behrens, 2016). The aim of the COPMAR project was to measure the outflow of HCHO and CHOCHO from the African Continent in order to validate both satellite measurements and model simulations using a Multi-AXis Differential Absorption Spectrometer (MAX-DOAS), which was installed on board of the research vessel with the telescope unit oriented perpendicularly to the heading, pointing towards the African continent.

The weather and viewing conditions during the cruise are summarised in Table 1. On all days of the cruise, clouds were observed with varying cloud fractions, heights and layers. However, only on 5 days, rain and poor viewing conditions were observed (14th, 15th, 18th, 19th and 24th October 2016).

## 3   Observations, modelling and data analysis

### 3.1   MAX-DOAS measurements and instrument set-up on the vessel

The Multi-AXis Differential Absorption Spectrometer (MAX-DOAS) instrument used in this study consisted of an Avantes spectrometer (AvaSpec-ULS2048x64) with a wavelength range from 288 nm to 500 nm, a spectral resolution of 0.6 nm Full Width at Half Maximum (FWHM), and a telescope unit which was installed on board of the research vessel (RV) Maria S. Merian. The instrument was continuously scanning a vertical plane pointing portside perpendicular to the vessel's direction, in order to generally point towards the African continent. Each scan lasted about 10 minutes. In the vertical plane, elevation angles from -3° to 8° were measured in 1° steps as well as, 10°, 15°, 30°, and 90°. A maximum exposure time of 0.1 seconds was used in order to be able to account for the ship's movement (roll angle) during data analysis. After correcting the angles

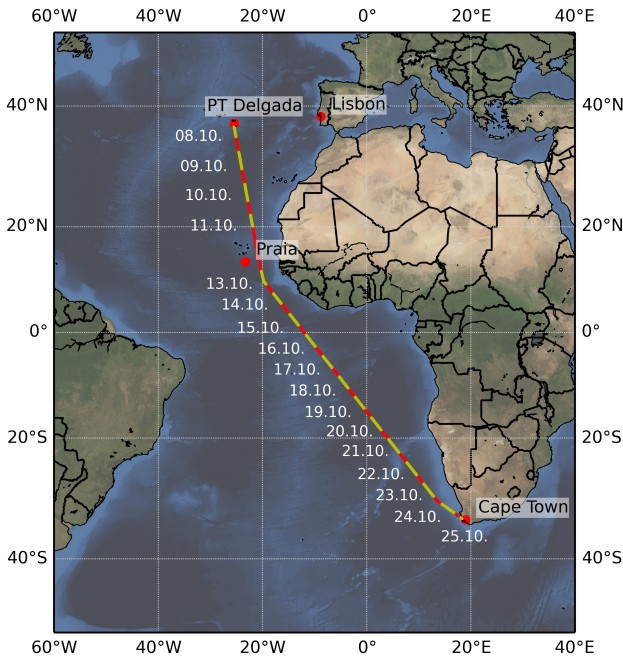

**Figure 2.** Cruise track (red) of MSM58/2 from Ponta Delgada (Azores) to Cape Town (South Africa). Times when the instrument was measuring are marked in yellow. On 12 October 2016, no measurements could be taken.

for the roll of the ship, the individual measurements within a 30 seconds period were sorted by viewing angle and binned into 1° intervals. Zenith sky measurements were averaged over 60 seconds. For the analysis, elevation angles from 0° to 90° above the horizon were used. Observations were taken at solar zenith angles (SZA) smaller than 96° for zenith sky measurements and for the off-axis at SZA smaller than 90°.

The ship sailed at nearly constant speed over ground with $\sim 12 - 14$ kn. The absolute wind direction was primarily from south. Consequently, headwinds were dominant during the campaign, shifting either side of the bow (Fig. 3). Measurements with a relative wind direction between 90° and 270° (grey shaded wind directions, Fig. 3) are excluded from the analysis, because of possible contamination by the vessel's plume. Furthermore, an intensity filter was used to exclude poor viewing conditions during heavy rain events (i.e. very dark scenes) to avoid the entailed high uncertainties.

## 3.2 Satellite measurements

The measurement data obtained during the COPMAR project are compared with satellite measurements of absorber vertical column densities (VCDs) from the OMI (Levelt et al., 2006) and GOME-2A/B (Callies et al., 2000; Munro et al., 2016) instruments.

For the comparison with stratospheric $NO_2$, data from the satellite instruments GOME-2A and GOME-2B are used. To retrieve the latitudinal dependency of stratospheric $NO_2$, a monthly mean for all measurements having SZA smaller than

**Table 1.** Weather and viewing conditions during COPMAR.

| Time period | Weather Conditions |
| --- | --- |
| 08. – 09.10.2016 | mostly cloudy |
| 10.10.2016 | decreasing cloud fraction |
| 11.10.2016 | increasing cloud fraction |
| 12.10.2016 | no measurements |
| 13.10.2016 | partly cloudy, different layers |
| 14.10.2016 | mostly cloudy, afternoon: cloudy, rain |
| 15.10.2016 | cloudy, rain |
| 16. – 17.10.2016 | increasingly cumulus clouds |
| 18. – 19.10.2016 | mostly cloudy, rain |
| 20.10.2016 | mostly cloudy, different layers |
| 21. – 22.10.2016 | morning: cloudy; afternoon: decreasing cloud fraction; evening: increasing cloud fraction |
| 23.10.2016 | partly cloudy |
| 24.10.2016 | morning: cloudy, rain, decreasing cloud fraction (mostly high clouds) |
| 25.10.2016 | nearly cloud free |

$85°$ is calculated for October 2016 using the DOAS fit settings from Richter et al. (2011). The satellite data are averaged over longitudes between $10°$ W and $40°$ W and divided by stratospheric air mass factors (AMFs). Additionally, satellite observations
for HCHO and CHOCHO are taken into account which are described in Alvarado et al. (2018) and Alvarado et al. (2019).

The measurement accuracy for satellite data are described in Boersma et al. (2004) and Lorente et al. (2017) as well as in Alvarado et al. (2014, 2019). Generally, over remote ocean areas the uncertainty of tropospheric columns is dominated by errors in the stratospheric column (for $NO_2$) and errors in the spectral fitting (Boersma et al., 2004). Boersma et al. (2004) found a relative uncertainty of up to $100\%$ for $NO_2$ in these areas. Thus, the satellite measurements over remote oceanic
areas are close to the detection limit, making it impossible to compare single satellite measurements to the COPMAR results. Consequently, monthly means are used for the comparison. For spatial collocation of the measurements, the satellite pixels are averaged within a $200\,km$ radius of the mid-day position of the ship.

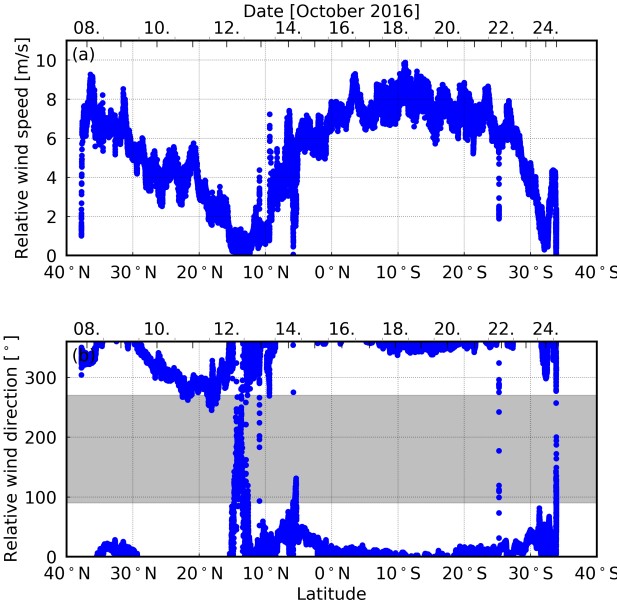

**Figure 3.** Relative (a) wind speed and (b) wind direction (0° are headwinds) during the campaign COPMAR. Grey shaded measurements are excluded from the analysis in order to avoid possible contamination by the vessel's plume.

## 3.3 AERONET data

The Maritime Aerosol Network (MAN; Smirnov et al., 2009) is a part of the AErosol RObotic NETwork (AERONET) project
(Holben et al., 1998) and provides ship-borne aerosol optical depth (AOD) measurements from Microtops sun photometers. These are handheld direct sun measuring devices which are separated into five spectral channels (Smirnov et al., 2009). In the dataset of the cruise MSM58/2, AODs at 380 nm, 440 nm, 500 nm, 675 nm, and 870 nm are available. The Ångström-exponent is calculated from the channels at 440 nm and 870 nm (Smirnov et al., 2009). Details of the Microtops handheld Sun photometers and their uncertainties can be found in Ichoku et al. (2002), Morys et al. (2001), Porter et al. (2001) and
Knobelspiesse et al. (2003, 2004). In the following, the AOD at 440 nm and the Ångström-exponent are used for comparison with our data.

## 3.4 MOZART-4 CTM

The 4-D fields of HCHO and CHOCHO concentrations are needed as a priori information for the calculation of VCDs for both MAX-DOAS and satellite data (Sect. 3.7). These a priori information have been taken from MOZART-4 (Emmons et al.,
2010) model output (available at https://www.acom.ucar.edu/wrf-chem/mozart.shtml). The model data have 6-hourly temporal and ∼ 1.9° × 2.5° horizontal resolution. We further used these model data for the interpretation of our results (Sect. 4.2 and Sect. 4.3). For the comparison, the model data are linearly interpolated in time on the cruise track.

### 3.5 FLEXPART

The FLEXible PARTicle dispersion model (FLEXPART) is a Lagrangian dispersion and transport model which can be used to simulate atmospheric transport of air parcels (https://www.flexpart.eu/; Stohl et al., 1998, 2005; Stohl and Thomson, 1999). Here, backward in time simulations (Seibert and Frank, 2004) are used to determine potential source regions of trace gas enhancements which were observed during three days during the cruise. The simulations were driven by ECMWF IFS (version CY41R2) wind fields at a horizontal resolution of $0.2°$ and a temporal resolution of 1 h. Backward simulations started at altitudes 20, 500, 1000, 1500, 2000, 2500, 3000, 3500, 4000, 4500, and 5000 m above the ship's location at hourly intervals between $\sim 11$ a.m. and $\sim 3$ p.m., coinciding with the measurement times. For each model run, 2 million individual air parcels were followed backwards in time for $2-4$ days; the simulated air tracer did not undergo any deposition or other loss processes.

### 3.6 FINN

The Fire INventory from NCAR (National Center for Atmospheric Research; FINN) is a dataset providing daily global emissions of trace gases and particles from biomass burning at 1 km resolution (Wiedinmyer et al., 2011). For this inventory, satellite observations of active fires from MODIS are used. The emissions are calculated by using land cover type, emission factors, and the estimated fuel loadings. In our analysis, we consider fires on the days before and after the day with high FLEXPART emission sensitivity to the surface, in order to account for uncertainties in fire detection.

### 3.7 DOAS analysis

The spectra measured by the Avantes instrument (see Sect. 3.1) have been analysed using the Differential Optical Absorption Spectroscopy (DOAS; Platt and Stutz, 2008; Burrows et al., 2011) technique which is based on Lambert Beer's law and describes the spectral attenuation of the initial intensity ($I_0$) of light due to extinction along the light path $s$:

$$I(\lambda, s) = I_0 exp(-\sigma(\lambda)\rho s). \tag{1}$$

Here $\lambda$ is the wavelength, $\sigma$ the absorption cross section, $I$ the reduced intensity, and $\rho$ the absorber concentration. This method can be used for ground-based as well as for satellite data, and for wavelengths in the ultra violet and visible spectral ranges. The main result of the DOAS analysis is the integrated concentration of trace gases $\rho(s)$ along the effective light path $s$, the so-called slant column density (SCD; Platt and Stutz, 2008; Burrows et al., 2011):

$$\text{SCD} = \int \rho(s) ds. \tag{2}$$

This quantity depends on the light path which is influenced by the SZA, the relative azimuth angle, the viewing direction of the instrument, and the viewing conditions. Therefore, a rough estimation of the absorber altitude is possible, using SCDs from different elevation angles due to the scan angle dependency of the light path. For pollution close to the ground, the highest SCDs are expected for the lowest elevation angle whereas for absorbers in elevated layers, the highest SCDs are expected at a higher elevation angle. For better interpretation, the SCDs are converted into VCDs, which are the integral of the trace gas

concentration from the surface to the top of the atmosphere (TOA) along the altitude $z$ (Platt and Stutz, 2008; Burrows et al., 2011):

$$\text{VCD} = \int_0^{TOA} \rho(z)dz. \tag{3}$$

For MAX-DOAS measurements, $I_0$ is usually a zenith sky measurement. Therefore, the derived quantities are differential values (differential SCD, dSCD), which has to be considered in the analysis. The sensitivity of the measurement to an absorber varies with altitude; this is expressed by the so-called box air mass factor (BAMF; Burrows et al., 2011), which is defined as $\text{BAMF}_i = \text{SCD}_i/\text{VCD}_i$ for an atmospheric layer $i$. BAMFs are calculated by radiative transfer models (here SCIATRAN, Rozanov et al., 2014), which take into account the viewing geometry and environmental effects (Platt and Stutz, 2008). The total AMF can be retrieved by using the BAMF and the trace gas concentration in the individual layers:

$$\text{AMF} = \sum_{i=0}^{TOA} \text{BAMF}_i \times \frac{\text{VCD}_i}{\text{VCD}}. \tag{4}$$

$\text{VCD}_i$ is the vertical distribution of the trace gases taken from the MOZART-4 model. The differential AMF (dAMF) is the difference between the AMF of the individual measurement and the AMF of the zenith sky reference measurement ($\text{dAMF} = \text{AMF}_{\text{meas}} - \text{AMF}_{\text{ref}}$). By applying the dAMF, the dSCD can be converted to VCD:

$$\text{VCD} = \frac{\text{dSCD}}{\text{dAMF}}. \tag{5}$$

### 3.8 DOAS-fit settings

#### 3.8.1 Factors influencing MAX-DOAS retrievals

The results of the DOAS analysis can be influenced by different effects. The instrument was mounted on the RV Maria S. Merian and therefore, a correction of the scan angle for the ship's roll angle is applied. This can lead to a mispointing of the scan angle. Because, ship movement data at high resolution are used for the correction, this error is minimised.

The choice of reference spectrum is also important for the detection of trace gases. There are different options for reference spectra, for example: a sequential zenith sky reference spectrum, a daily noon zenith sky reference spectrum, or a fixed zenith sky reference spectrum for the full campaign. In this study, the trace gases are expected to be in elevated layers. Furthermore, satellite measurements indicate that they depend on latitude (see Fig. 1). For both cases, a fixed reference spectrum is needed as otherwise the trace gas signal might not be observable because elevated layers contribute to both horizon and zenith-sky measurements in a similar way. Nevertheless, this choice does not impair the detectability of pollution located in the boundary layer. An important criterion for the choice of the fixed reference spectrum is that this spectrum is not affected by the elevated layer.

Each MAX-DOAS measurement is influenced by a systematic and a random error (Irie et al., 2011; Takashima et al., 2012). The random error can be estimated from the residual of the fitted trace gases (Takashima et al., 2012). However, also

systematic errors can be found in the residual such as not perfectly resolved cross sections. Furthermore, Boersma et al. (2004) and Takashima et al. (2012) showed that the temperature dependency of the cross section also leads to systematic errors in the retrieved SCD. For example for $NO_2$, it is in the order of $0.4\,\%\,K^{-1}$ (Takashima et al., 2012). For the vertical columns, the errors are in the order of 15 to 20 % (Takashima et al., 2012; Peters et al., 2012).

The measurements were taken at low latitudes over the ocean which leads to strong water vapour absorption. This strong absorption, in combination with a not perfectly resolved water vapour cross section, can introduce problems in the fitting procedure. Therefore, the mean residual is used as an additional pseudo cross section for CHOCHO retrievals (see Sect. 3.8.4). Furthermore, the strong $H_2O$ absorption could contribute to interferences with weak absorbers. In this case, the differences should be reduced/enhanced for higher/lower elevation angles, respectively. However, this is not observed (compare Sect. 4).

The detection limit for the DOAS-method for ship-based measurements depends on the trace gas and on the instrument. In this study, the detection limit is calculated for each trace gas individually with the method from Sinreich et al. (2010). They analysed CHOCHO as well and defined the detection limit as twice the root mean square (RMS) of the fit residual divided by the strongest peak of the convoluted differential cross section. For the three analysed trace gases, the resulting detection limits are discussed in Sects 3.8.2 to 3.8.4.

### 3.8.2 Stratospheric $NO_2$

For the evaluation of the instrument setup, stratospheric $NO_2$ is used, which is well known from previous studies such as Peters et al. (2012), Kreher et al. (1995), and Senne et al. (1996). For this $NO_2$ retrieval, a fitting window from 450 to 490 nm is used. Absorption cross sections for $NO_2$ (Vandaele et al., 1998) at 220 K and orthogonalized to this a cross section at 298 K are used to account for the stratosphere and the troposphere, respectively. Furthermore, an $O_3$ (Serdyuchenko et al., 2014; Gorshelev et al., 2014) cross section at 223 K with an $I_0$ correction of $10^{20}$ molec $cm^{-2}$ (Platt et al., 1997; Richter, 1997) are included in the fit. Additionally, cross sections for $O_4$ (Thalman and Volkamer, 2013), $H_2O$ (Rothman et al., 2010), and a cross section to correct the Ring effect (Vountas et al., 1998) are included in the analysis, along with a 5th order polynomial. Figure 4 shows an example fit for stratospheric $NO_2$.

A fixed reference spectrum measured on 17 October 2016 at 12:20 UTC and 4.03° SZA is used to analyse the data. On this day, the ship was close to the Equator ($\sim 9.6°$ S) with negligible tropospheric $NO_2$ (known from MAX-DOAS measurements analysed with a sequential zenith sky reference spectrum and satellite observations) and good weather conditions. The retrieved dSCD for $NO_2$ are converted into VCD by using AMF calculated at a wavelength of 470 nm. To analyse stratospheric $NO_2$, only zenith sky measurements with a SZA smaller than 92° are used.

For the conversion into total VCD, a reference VCD ($VCD_{ref}$) is calculated as follows:

$$VCD_{ref} = \frac{VCD_{a.m.} + VCD_{p.m.}}{2} = \frac{dSCD_{a.m.} + dSCD_{p.m.}}{2 \times (AMF_{88°} - AMF_{ref})} \approx 1.97 \times 10^{15} \frac{molec}{cm^2} \tag{6}$$

with the assumption, that the reference spectrum is at noon and $VCD_{ref}$ is the mean value of the morning ($VCD_{a.m.}$) and evening value ($VCD_{p.m.}$), both taken at SZA $= 88°$. The AMFs are calculated by using the radiative transfer model SCIATRAN (Rozanov et al., 2014): $AMF_{ref} = AMF(10°) = 1.11$ and $AMF(88°) = 12.99$. Afterwards, a reference SCD ($SDC_{ref}$) can be

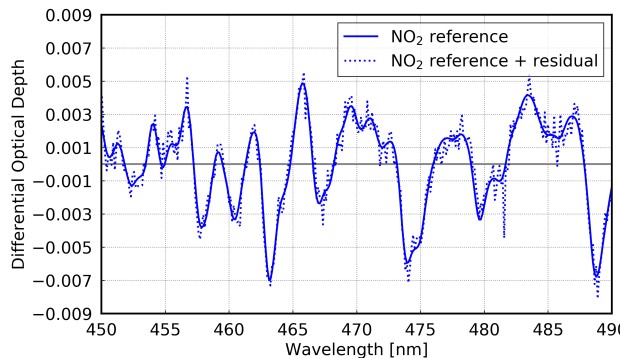

**Figure 4.** Example of a stratospheric $NO_2$ fit for 14 October 2016 at 6:58 UTC (latitude: $\sim 7.4°$ N; longitude: $\sim 17.8°$ W). Both are shown, the reference scaled with the SCD as well as the scaled reference plus the residual. Zenith sky measurement with a SCD of $3.99 \times 10^{16}$ molec cm$^{-2}$ and a RMS of $6.9 \times 10^{-4}$ at 90.9° SZA.

calculated:

$$\mathrm{SCD_{ref}} = \mathrm{AMF_{ref}} \times \mathrm{VCD_{ref}} \approx 2.14 \times 10^{15} \frac{\mathrm{molec}}{\mathrm{cm}^2} \tag{7}$$

which is added to the dSCD to retrieve total SCD:

$$\mathrm{SCD} = \mathrm{dSCD} + \mathrm{SCD_{ref}}. \tag{8}$$

After the conversion into total VCDs, the latitudinal dependency and the diurnal cycle of stratospheric $NO_2$ are calculated. The longest light path in the stratosphere and thus the highest sensitivity to stratospheric $NO_2$ can be found during twilight. Therefore, measurements between 88° and 92° SZA are used for both morning and evening measurements for the analysis of the latitudinal dependency of stratospheric $NO_2$. For the calculation of the stratospheric diurnal cycle, half hour binned values are calculated, because of the higher signal to noise ratio compared to the single measurements.

The observed stratospheric $NO_2$ values are well above the detection limit. The mean RMS is $3.5 \times 10^{-4}$ and the strongest convoluted differential peak of the $NO_2$ cross section $(1.4 \times 10^{-19})$ which leads to an estimated detection limit of $4.8 \times 10^{15}$ molec cm$^{-2}$ for dSCDs (Sect. 3.7). For large SZAs with an AMF of around 15, the detection limit for VCDs is therefore $3.2 \times 10^{14}$ molec cm$^{-2}$.

### 3.8.3 Formaldehyde

For the HCHO retrieval, a fitting window from 336.5 to 359 nm is used. Absorption cross sections for HCHO at 297 K (Meller and Moortgat, 2000), $O_3$ (Serdyuchenko et al., 2014) at 223 K and orthogonalised 243 K, as well as $NO_2$ at 298 K (Vandaele et al., 1998) are used. For the latter two gases, an $I_0$ correction (Aliwell et al., 2002) of $10^{20}$ molec cm$^{-2}$ and $10^{17}$ molec cm$^{-2}$ is applied, respectively. Additional cross sections are $O_4$ (Thalman and Volkamer, 2013), BrO at 223 K (Fleischmann et al., 2004), as well as a cross section to correct the Ring effect (Vountas et al., 1998) and a polynomial of degree 5. Furthermore, the

mean residual of the whole cruise is included as additional cross sections to improve the fit results. By including this additional cross section, the fit RMS reduces by approximately 50 % (not shown). Figure 5 shows an example fit for HCHO.

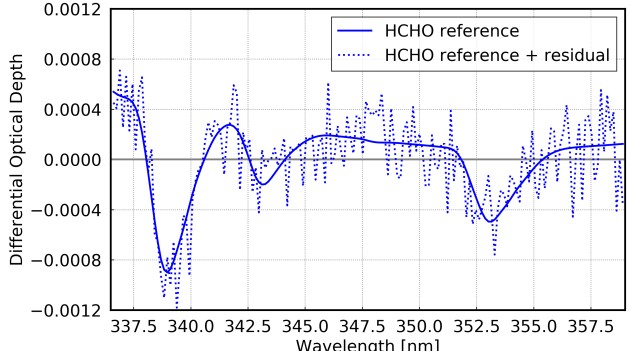

**Figure 5.** Example of a HCHO fit for the 13 October 2016 at 14:38 (latitude: $\sim 10.1°$ N; longitude: $\sim 19.8°$ W). The SCD is $2.52 \times 10^{16}$ molec cm$^{-2}$ with a RMS of $2.2 \times 10^{-4}$ at $29.46°$ SZA and a viewing angle of $15°$.

As for the case of stratospheric $NO_2$, also for HCHO a fixed reference spectrum is used. The reference spectrum is from 23 October 2016 ($\sim 29.2°$ S) around noon. This date was chosen because of low HCHO content in the overhead column (expected from satellite observations), whereas closer to the Equator, higher HCHO values would be expected. For the conversion to VCDs, dAMF were calculated at the wavelength 338 nm as shown in Eq. 5.

Our measurements show HCHO dSCDs consistently above the detection limit. The measurements have a mean RMS of $2.9 \times 10^{-4}$, which leads to a detection limit of $\sim 2.2 \times 10^{15}$ molec cm$^{-2}$ for daily mean dSCDs, depending on the number of measurements per day. Applying a dAMF of 1.1 (representative for $30°$ elevation angle; see also Sect. 3.7), this translates to a detection limit of $\sim 1.9 \times 10^{15}$ molec cm$^{-2}$ in the VCDs.

For the analysis, only measurements at SZA $< 70°$ are used, because for larger SZA the measurement uncertainty increases. HCHO has a global background due to the oxidation of $CH_4$ (Sect. 1). Therefore, the overhead column from the reference measurement needs to be considered for the conversion into total VCDs which is done by adding the simulated column ($VCD_{Model} = 3.7 \times 10^{15}$ molec cm$^{-2}$) from the MOZART-4 data at the location of the reference measurement to the VCD:

$$VCD_{total} = VCD + VCD_{Model}. \tag{9}$$

### 3.8.4 Glyoxal

The CHOCHO fitting window is from 433 to 460 nm with a polynomial degree of 4. Absorption cross sections are CHOCHO (Volkamer et al., 2005) and $O_3$ (Bogumil et al., 2003) at 223 K with an $I_0$ correction of $10^{20}$ molec cm$^{-2}$. Furthermore, $NO_2$ cross sections (Vandaele et al., 1998) at 298 K with $I_0$ correction of $10^{17}$ molec cm$^{-2}$ and at 220 K orthogonalised to 298 K, an $O_4$ cross section (Thalman and Volkamer, 2013) at 293 K, a $H_2O$ cross section (Rothman et al., 2010) at 296 K with an $I_0$ correction of $10^{24}$ molec cm$^{-2}$ as well as a cross section to correct for the Ring effect (Vountas et al., 1998) are included in the

275 fit. The fit residual shows structures which are related to strong water vapour absorption over the ocean and possible missing peaks in the water cross section (not shown).

To improve the fit residual, two additional cross sections were tested, an alternative $H_2O$ cross section (Polyansky et al., 2018) and a mean residual as an additional cross section. The alternative $H_2O$ cross section did not reduce the fit residual, and therefore, it is not used here. The mean residual was calculated for all measurements during the cruise and the fit was repeated
including this mean residual as additional pseudo-absorber. This additional cross section clearly improved the fit residual, and therefore, this pseudo-absorber is used in the following. Figure 6 shows an example fit for CHOCHO.

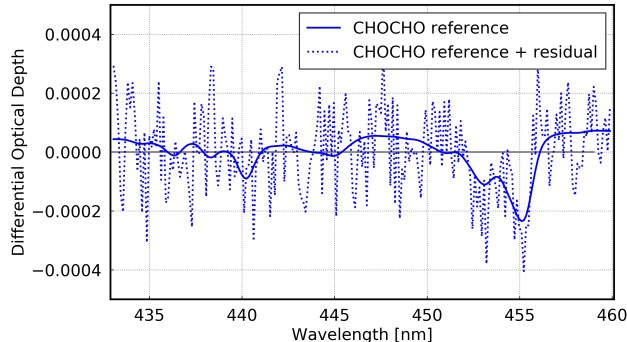

**Figure 6.** Example of a CHOCHO fit for the 13 October 2016 at 11:33 (latitude: $\sim 10.8°$ N; longitude: $\sim 20.0°$ W). The SCD is $6.17 \times 10^{14}$ molec cm$^{-2}$ with a RMS of $1.3 \times 10^{-4}$ at 29.75° SZA and a viewing angle of 7°.

As for HCHO, a reference spectrum from the 23 October 2016 around noon was chosen. Also for CHOCHO, low overhead columns are expected in this area. Since background concentrations of CHOCHO are not expected, no global background value has to be added. For the conversion into VCDs, AMFs were calculated at the wavelength 433 nm (see Eq. 5), and only
285 measurements at SZA $< 70°$ are used in order to limit noise occurring at low intensities.

Our analysis of the MAX-DOAS measurements yields CHOCHO dSCDs which are often below the detection limit. The RMS is in the order of $1.6 \times 10^{-4}$, resulting in a detection limit of $\sim 1.7 \times 10^{14}$ molec cm$^{-2}$ for daily averages of the dSCDs, depending on the number of measurements (Fig. 11). Applying a dAMF of 1.2 (representative for 30° elevation angle; see also Sect. 3.7), this translates to a detection limit of $\sim 1.4 \times 10^{14}$ molec cm$^{-2}$ in the VCDs.

# 4 Results

## 4.1 Evaluation of stratospheric $NO_2$

To demonstrate the performance of the instrument setup, we analysed the diurnal and latitudinal variation of stratospheric $NO_2$ and compared our results with satellite data and previous studies. The diurnal cycle of stratospheric $NO_2$ is closely related to the photolysis of the reservoir species $N_2O_5$ (Solomon et al., 1986). During daylight, $NO_2$ increases as result of $N_2O_5$
photolysis whereas at night, $N_2O_5$ increases and $NO_2$ is removed. Thus, it has a minimum in the morning, followed by a linear

increase. As shown by Peters et al. (2012), this diurnal cycle can be observed over remote oceanic areas, where the stratospheric signal is usually not impaired by the presence of tropospheric $NO_2$. On the cruise MSM58/2, such a diurnal cycle could also be observed which is exemplarily shown for 15 October 2016 in Fig. 7. On that day, the linear increase during daytime amounted to $7.31 \times 10^{13}$ molec cm$^{-2}$ h$^{-1}$ ($\sim 2.5°$ N; $\sim 13.9°$ W). Previous studies found similar results, e.g., Peters et al. (2012) with an increase of $8.7 \times 10^{13}$ molec cm$^{-2}$ h$^{-1}$ for the tropics and Gil et al. (2008) with an increase of $6 \times 10^{13}$ molec cm$^{-2}$ h$^{-1}$ for the subtropics.

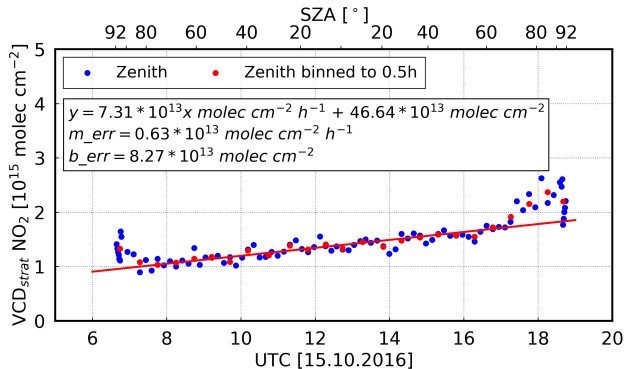

**Figure 7.** The diurnal cycle of stratospheric $NO_2$ (15.10.2016; latitude: $\sim 2.5°$ N; longitude: $\sim 13.9°$ W). In red, the data are binned to 0.5 h values to improve the signal to noise. Also shown is the regression line which represents a linear increase during the day. The data are shown for SZAs smaller than 92°. Furthermore, the error of the slope ($m_{err}$) and of the intercept ($b_{err}$) are presented.

The diurnal cycle of stratospheric $NO_2$ depends on the latitude, with a smaller increase in the tropics and a stronger increase towards the mid-latitudes, as shown in Fig. 8. The presented morning and evening MAX-DOAS values (averaged over all measurements with SZAs between 88° and 92°) approximately correspond to the first and last red dots in Fig. 7, respectively. They both show a local minimum near the Equator and increase towards the mid-latitudes.

The morning MAX-DOAS values range from $2.0 \times 10^{15}$ molec cm$^{-2}$ at 32.8° N to $1.2 \times 10^{15}$ molec cm$^{-2}$ close to the Equator and increase again in the Southern Hemisphere, to $2.3 \times 10^{15}$ molec cm$^{-2}$ at 33.7° S. On the other hand, the evening MAX-DOAS values are higher ($3.4 \times 10^{15}$ molec cm$^{-2}$ at 35.4° N to $2.0 \times 10^{15}$ molec cm$^{-2}$ close to the Equator and $3.5 \times 10^{15}$ molec cm$^{-2}$ at 32.8° S). This "U"-shape can also be observed in the satellite measurements. Given the 09:30 local time (which corresponds to $\sim 11$ UTC) overpass time of both GOME-2 instruments, one would expect that the satellite values are close to the morning MAX-DOAS values. Taking measurement uncertainties into account, this behaviour can indeed be found for the GOME-2B data. However, the GOME-2A measurements are slightly higher than the observed morning MAX-DOAS values, which is probably related to the degradation of the instrument (Dikty and Richter, 2011). The uncertainties for the retrieved MAX-DOAS measurements are calculated by error propagation using the fitting error and the assumed AMF uncertainty of $\pm 1$. The uncertainty in VCD$_{ref}$ has been neglected (an assumed uncertainty of 30 % in VCD$_{ref}$ amounts to a value of $0.59 \times 10^{15}$ molec cm$^{-2}$ which corresponds to an uncertainty of $0.65 \times 10^{15}$ molec cm$^{-2}$ for the SCD$_{ref}$, resulting in a negligibly small uncertainty of $0.05 \times 10^{15}$ molec cm$^{-2}$ for twilight measurements).

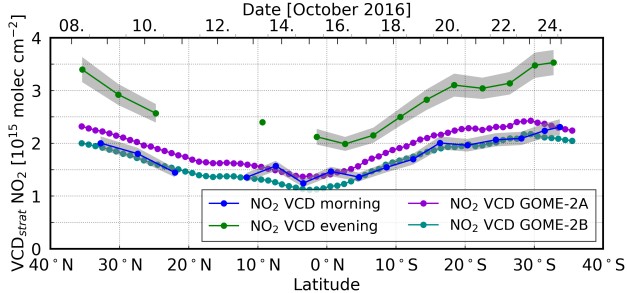

**Figure 8.** The latitudinal dependency (from North to South) of stratospheric $NO_2$ over the Atlantic Ocean shows a local minimum near the Equator and an increase towards the mid-latitudes. MAX-DOAS morning and evening values have been averaged over all zenith sky measurements for SZAs between $88°$ and $92°$. The colour shaded areas indicate the errors of the measurements.

Our results for stratospheric $NO_2$ are in good agreement with previous studies from Kreher et al. (1995) and Senne et al. (1996), especially in the Northern Hemisphere and across the Equator. In the Southern Hemisphere (at $30°$ S) however, the values reported by Kreher et al. (1995) and Senne et al. (1996) are significantly higher than our measurements presented here, with values of approximately $3 \times 10^{15}$ molec cm$^{-2}$ (morning) and $5 \times 10^{15}$ molec cm$^{-2}$ (evening). This could be related to either intra-annual (our measurements were taken a few weeks earlier in the year compared to the two previous studies) or inter-annual variability. Kreher et al. (1995) and Senne et al. (1996) analysed measurements from 1990 and 1993, respectively. These measurements differ by approximately $1 \times 10^{15}$ molec cm$^{-2}$ at $30°$ N. Furthermore, Kreher et al. (1995) showed that the values differ for higher latitudes between the individual years. In the Southern Hemisphere, we observe a stronger increase towards the mid-latitudes compared to the Northern Hemisphere, consistent with the findings by Kreher et al. (1995) and, for the Pacific Ocean, by Peters et al. (2012).

Overall, the instrument performed well as the diurnal cycle and the latitudinal dependency of stratospheric $NO_2$ are clearly visible in our measurements and our results agree with previous studies as well as satellite data. Therefore, we are confident in the suitability of our measurements to analyse the weak absorbers HCHO and CHOCHO.

### 4.2 Formaldehyde

The latitudinal variation of daily mean MAX-DOAS HCHO dSCDs observed during the cruise shows enhanced values in an elevated layer at both $\sim 10°$ N and $\sim 5°$ S. This coincides with the area of expected outflow from the African continent as seen in satellite measurements (Fig. 1 a). Typically, the HCHO concentration in polluted areas is expected to be highest close to the surface (De Smedt et al., 2008; Heckel et al., 2005) due to the primary emission sources of HCHO and rapid photochemical production from VOC precursors. Thus, also the highest dSCDs should be observed at low elevation angles near the sources. Figure 9 illustrates the daily mean HCHO dSCDs during the cruise at different elevation angles. During the cruise, whenever low HCHO columns ($\sim 0.5 \times 10^{16}$ molec cm$^{-2}$ for zenith sky measurements) are observed, the dSCDs are indeed highest at the lowest elevation angles.

Around 10° N and 5° S (on 13 October, 14 October, and 17 October, respectively), we observe enhanced HCHO values ($> 1 \times 10^{16}$ molec cm$^{-2}$ for the off-axis measurements). Here, the dSCD are largest at higher elevation angles, which indicates that HCHO is predominantly located in an elevated layer. Still, the zenith sky measurements show the lowest values, but the other elevation angles are ordered differently than usual, with highest HCHO dSCD at elevation angles between $8 - 15°$. This pattern is not only visible in the daily mean as depicted in Fig. 9, but can also be observed for the individual vertical scans (not

shown here).

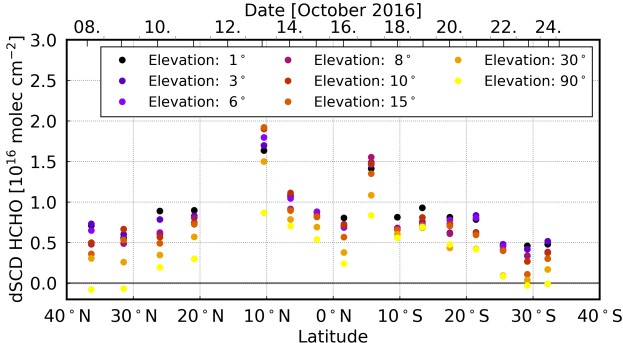

**Figure 9.** Daily mean HCHO dSCDs over the Atlantic Ocean over the course of the cruise (from North to South). Enhanced values in the areas of expected outflow show a different scan angle dependency.

When converting the measured HCHO dSCD to VCDs (using the 30° elevation angle and HCHO profiles from the MOZART-4 model), the values on 13 October, 14 October, and 17 October remain clearly enhanced (see Fig. 10 a). At least the first region sampled on 13/14 October ($\sim 10°$ N) indeed coincides with the area for which satellite observations regularly show enhanced HCHO values (compare Fig. 1 a and Fig. 2). On these days, also MOZART-4 data are enhanced in the region.

However, differences between the datasets are visible. On 13 October, MAX-DOAS and MOZART-4 data are higher than the satellite data. While on 14 October, MAX-DOAS and satellite data are slightly smaller than on 13 October, MOZART-4 data are further increased. On 17 October (5° S), there is no clear sign of a HCHO enhancement in the satellite and model data in contrast to MAX-DOAS measurements. These differences could be related to several reasons. The MAX-DOAS data are measured on single days, whereas the satellite datasets are monthly means and MOZART-4 time series are interpolated on the

cruise track. Thus, the differences between MAX-DOAS and satellite data could be explained by single, isolated outflow events on specific days, which are not distinguishable from background values in the monthly averaged satellite data. The interpolation of the model data could lead to differences between model and MAX-DOAS data as for example isolated events might not be represented in the model data. Furthermore, the satellite and model data are averages over larger areas which would dilute the magnitude of the measured or simulated values. Thus, comparing HCHO columns retrieved from OMI and GOME-2B

radiances (Sect. 3.2) and integrated columns from simulated MOZART-4 profiles (Sect. 3.4; interpolated on the cruise track) to our MAX-DOAS VCDs, the results generally confirm the finding of enhanced HCHO satellite and model columns. The datasets show good agreements with correlation coefficients larger than 0.72 (Table 2).

The MAX-DOAS measurements are mostly higher than the satellite observations (Fig. 10 a) which results in a slope larger than one (1.91, OMI and 1.24, GOME-2B) and a large offset for GOME-2B ($3.05 \times 10^{15}$ molec cm$^{-2}$, GOME-2B and $0.01 \times 10^{15}$ molec cm$^{-2}$, OMI) of the regression line (Table 2). These differences are clearly visible between $20°$ N and $32°$ N (09 October – 11 October) and between $10°$ S and $22°$ S (18 October – 21 October; Fig. 10 a), which have been measured in clean remote ocean areas with low pollution.

Compared to the MOZART-4 model values, the MAX-DOAS observations are often higher which can also be observed in a low slope of 0.68. Close to the Equator, the values show good agreement in the area of expected pollution outflow which leads to a high correlation coefficient between the two datasets of 0.72. However, the offset is also high between the two datasets ($4.82 \times 10^{15}$ molec cm$^{-2}$).

Several reasons can contribute to an enhancement in MAX-DOAS data. Large differences between the MAX-DOAS and the satellite or model data are found over regions with low air pollution north and south of the Equator. In these areas, high measurement uncertainties can be found in the satellite data due to the low columns which might influence the retrieved satellite values. It is also possible that the model and satellite results underestimate the VCDs, because of the potentially localised nature of the enhancements (see also Sect. 5). Increased uncertainties in the MAX-DOAS data in this region can be ruled out as the DOAS fit RMS is nearly constant during the whole cruise (see Sect. 3.8.3). Furthermore, $H_2O$ interferences might contribute to the differences. In that case, the differences should be reduced/increased for higher/lower elevation angles, respectively. However, this can also be ruled out as a similar behaviour is visible for all viewing directions. Additionally, bad weather conditions can be ruled out, because the affected days of the MAX-DOAS measurements had different weather conditions (see Table 1) and an intensity filter was used to exclude poor viewing conditions (see Sect. 3.1). Similar conditions were also used for satellite values. Here, only measurements with geometric cloud fraction smaller 0.3 were included in the analysis.

The AMF could introduce the differences between the datasets. However, this seems unlikely as the MOZART-4 model is used for the AMF calculations for both MAX-DOAS and satellite measurements and this model does not show a similar behaviour to the MAX-DOAS measurements. For example, if the model underestimates/overestimates the total amount of HCHO in the atmosphere, this would in good approximation not change AMFs, and therefore, the VCDs of both datasets are not influenced. Thus, no difference would be introduced between the satellite and MAX-DOAS data. Possibly, the differences between the datasets can be related to the HCHO model profile if it would differ from the real atmospheric profile. On the one hand, the model could put the HCHO enhancement at the wrong altitude. This could lead to both an underestimation or an overestimation of the MAX-DOAS VCDs, depending on the profile and the SZA or relative azimuth angle, similar for satellite measurements. On the other hand, the model could miss an additional, elevated layer of transported HCHO or an additional layer close to the surface. During those days, the HCHO in the model is located at $\sim 3$ km altitude, where the MAX-DOAS sensitivity is highest. Therefore, an additional layer of HCHO would reduce the MAX-DOAS AMF, leading to a higher MAX-DOAS VCD; if located close to the surface, also the satellite VCD would increase. In this case, the difference between MAX-DOAS and satellite VCD would be reduced due to the stronger altitude dependence of the satellite BAMF compared to the MAX-DOAS BAMF. If the additional layer were located above the model elevated layer, the satellite AMF would increase due to the better visibility, leading to decreased satellite VCDs, further increasing the differences between satellite and MAX-

DOAS VCDs. Additionally, aerosols could influence the AMFs leading to differences in satellite and MAX-DOAS VCDs observations.

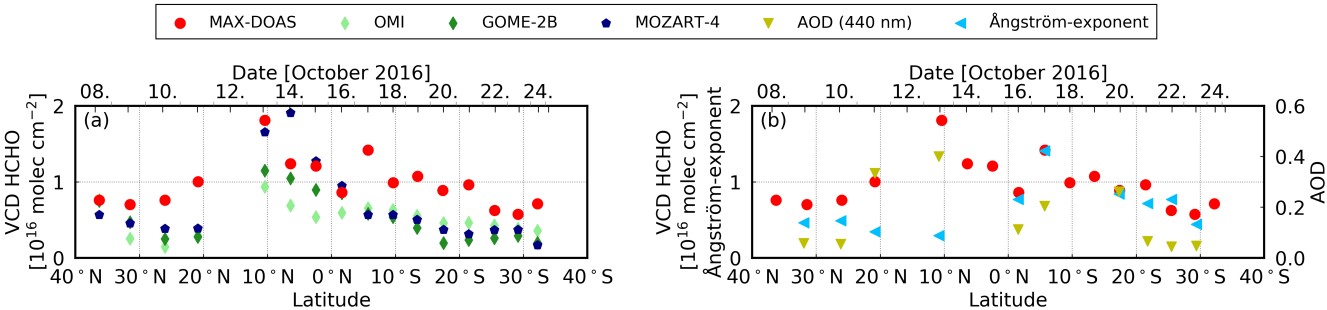

**Figure 10.** Daily mean HCHO VCDs over the Atlantic Ocean over the course of the cruise (from North to South). (a) Enhanced HCHO values can be observed in the area of expected outflow. Satellite observations and model values are also enhanced in this area. (b) AOD and Ångström-exponent from AERONET measurements are partly enhanced in these areas, depending on the source region.

**Table 2.** Correlation, slope and intercept between HCHO MAX-DOAS and satellite measurements as well as model simulations. The values in brackets are the standard errors for slope and intercept.

| | correlation coefficient | slope | intercept ($10^{15}$ molec cm$^{-2}$) |
|---|---|---|---|
| OMI vs MAX-DOAS | 0.79 | 1.91 (0.36) | 0.01 (1.84) |
| GOME-2B vs MAX-DOAS | 0.72 | 1.24 (0.26) | 3.05 (1.48) |
| MOZART-4 vs MAX-DOAS | 0.72 | 0.68 (0.15) | 4.82 (1.16) |

## 4.3 Glyoxal

The observed CHOCHO dSCDs are mostly higher at low elevation angles, again indicating that the trace gas is located close to the ground (Fig. 11). The measured dSCDs are mostly smaller than $0.2 \times 10^{15}$ molec cm$^{-2}$ fluctuating around zero. Thus, the measurements are below the calculated detection limit (see Sect. 3.7). Nevertheless, enhanced CHOCHO dSCD were observed on 13 and 14 October (around 10° N) as for HCHO, with a maximum value of about $\sim 0.30 \times 10^{15}$ molec cm$^{-2}$ for

15° and 6° elevation angle measurements (see Fig. 11). On these two days, the measurements are above the detection limit and the scan angle dependency is slightly different, which indicates that the observed CHOCHO is located in an elevated layer, albeit not as clearly as for HCHO. On some days, the measurements for lower elevation angles are above the detection limit (17 October and 21 October). Here, CHOCHO might be located closer to the ground.

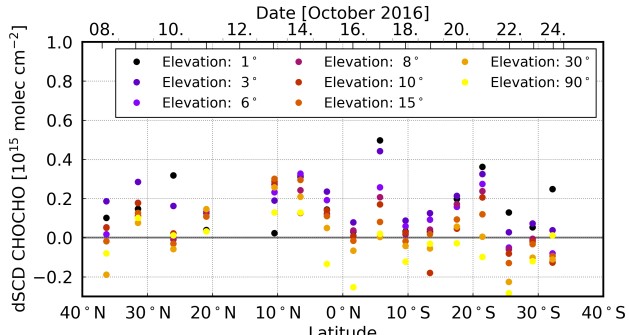

**Figure 11.** The latitudinal dependency (from North to South) of daily mean CHOCHO dSCDs over the Atlantic Ocean. The 13 and 14 October show a different scan angles dependency.

  When converting the measured CHOCHO dSCDs to VCDs (again using the 30° elevation angle and CHOCHO profiles
from the MOZART-4 model), the values on 13 and 14 October remain enhanced (see Fig. 12). This region indeed coincides with the area for which satellite observations regularly show enhanced CHOCHO columns (see Fig. 1 b). Comparing columns retrieved from OMI radiances and integrated from simulated MOZART-4 profiles (interpolated on the cruise track) to our MAX-DOAS VCDs, this generally confirms the finding of enhanced CHOCHO satellite and model columns. CHOCHO VCDs from all three datasets show enhanced values around 10° N while MAX-DOAS and MOZART-4 values further north
and further south are close to zero. In comparison, OMI observations show enhanced CHOCHO columns throughout the tropics in Fig. 12. This behaviour could be explained by an elevated CHOCHO layer which is not represented in the model and cannot be detected by MAX-DOAS measurements due to their low sensitivity in the free troposphere. The enhanced values throughout the tropics are also represented in the slope of the regression line with 0.32, nevertheless only a negligible offset of $-0.05 \times 10^{15}$ molec cm$^{-2}$ was found (Table 3). The correlation between MAX-DOAS and OMI CHOCHO VCDs is 0.56.
The modelled MOZART-4 data are mostly close to or slightly higher than the observed MAX-DOAS values; in particular, the observed CHOCHO enhancements are less pronounced in our measurements than in the model data resulting in a slope of the regression line of 1.06 with a high standard error of 0.29, whereas the offset is small ($-0.06 \times 10^{15}$ molec cm$^{-2}$). Overall, the MOZART-4 and MAX-DOAS CHOCHO data have a correlation coefficient of 0.55.

## 4.4 Aerosol

Around 10° N (13 October) the AERONET AOD (0.40) and the MAX-DOAS HCHO and CHOCHO are clearly enhanced in an elevated layer for the traces gases (Fig. 10 b and Fig. 12 b). Similar results for MAX-DOAS measurements were found

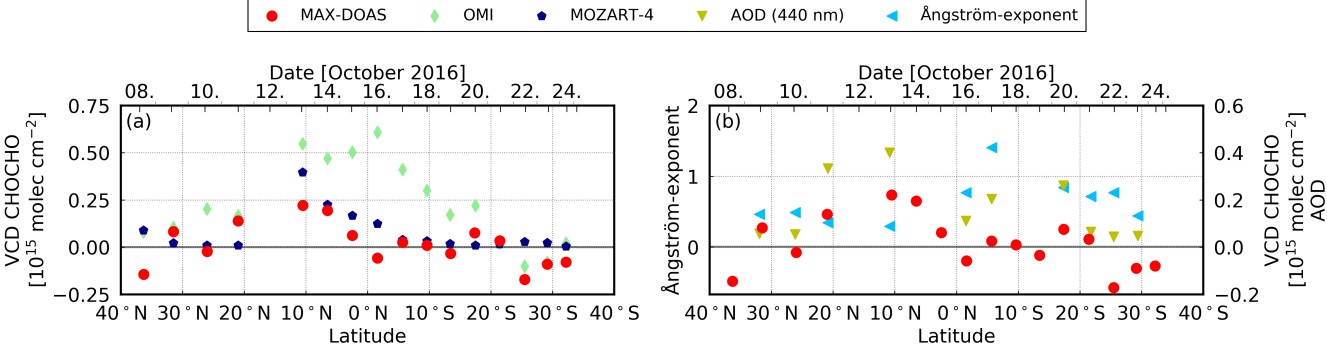

**Figure 12.** The latitudinal dependency (from North to South) of daily mean CHOCHO VCDs over the Atlantic Ocean. (a) Enhanced CHOCHO values can be observed in the area of expected outflow. As for HCHO, satellite observations and model values are enhanced in this area. (b) AOD from AERONET dataset is enhanced in this area. For this plot a similar axis as for Fig. 10 is used.

**Table 3.** Correlation, slope and intercept between CHOCHO MAX-DOAS and satellite measurements as well as model simulations. The values in brackets are the standard errors for slope and intercept.

|  | correlation coefficient | slope | intercept |
|---|---|---|---|
|  |  |  | ($10^{15}$ molec cm$^{-2}$) |
| OMI vs MAX-DOAS | 0.56 | 0.32 | -0.05 |
|  |  | (0.11) | (0.03) |
| MOZART-4 vs MAX-DOAS | 0.55 | 1.06 | -0.06 |
|  |  | (0.29) | (0.03) |

at 6° N (14 October), where no AOD observations are available. At approximately 20° N (11 October), the AERONET AOD shows also slightly enhanced values (0.33). For the MAX-DOAS measurements, the values are slightly increased for HCHO with the same scan angle dependency as expected for pollution close to the ground. Here, it is important to remember that
for CHOCHO the values are below/close to the detection limit. On both days (11 October and 13 October), the Ångström-exponent is in the order of 0.3. The classification of Toledano et al. (2007) shows that in this area the main aerosol type is desert dust which agrees with the findings of Takemura et al. (2000), Ridley et al. (2012), Ichoku et al. (2004), and Schepanski et al. (2017).

At approximately 5° S (17 October), the relation between MAX-DOAS measurements, AOD, and Ångström-exponent is
slightly different. The AOD is 0.20 with a high Ångström-exponent of 1.41 indicating a different source than in the Northern

Hemisphere. Here, the main aerosol type is on the edge between marine and continental aerosol, after the classification of Toledano et al. (2007). Because of the enhanced HCHO in an elevated layer, it might originate from continental sources.

At approximately 18° S (20 October), the AOD is increased with a value of 0.26 and the Ångström-exponent has a value of 0.84. However, HCHO and CHOCHO show no enhancement. On this day, a container vessel was sailing in front of RV Maria S. Merian and increased tropospheric $NO_2$ columns were observed with a clear scan angle dependency expected for pollution close to the surface (not shown). The main aerosol type classified after Toledano et al. (2007) is desert dust. The reason for the AOD enhancement might be that the air mass is strongly mixed with marine aerosols which were visible on the first days and the last days of the cruise. Thus, a mixture of exhaust and marine aerosols are observed potentially leading to the desert dust classification.

## 4.5  Model simulations

MOZART-4 model simulations (Emmons et al., 2010) show elevated layers of enhanced HCHO and CHOCHO concentrations between $\sim$ 3000 m to $\sim$ 6000 m (Fig. 13 and Fig. 14, respectively), on 13 October, 14 October, and 17 October. On these days, the scan angle dependency of the measured dSCDs suggests the presence of an elevated layer of HCHO and CHOCHO and of only HCHO, respectively. However, the model shows generally between 20° N and approximately 30° S increased values in an elevated layer, which cannot be confirmed by our measurements. On 13 October and 14 October, the model additionally shows enhanced VOC concentrations between the surface and approximately 2000 m altitude. This lower layer of high VOC concentrations could explain the overall higher dSCDs compared to the other days, probably indicating that the enhanced dSCDs measured on that day are only partly located in an elevated layer.

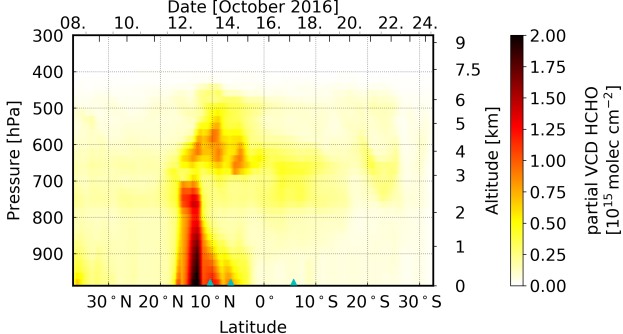

**Figure 13.** HCHO MOZART-4 profiles as used for VCDs calculations, interpolated on the cruise track (from North to South). The blue triangles on the bottom indicate the position of RV Maria S. Merian on the days with unusual scan angle dependency. For the calculation of the altitude, a mean temperature profile is used.

To further complement the interpretation of our results, we conducted backward simulations with the FLEXPART dispersion model (Stohl et al., 1998, 2005; Stohl and Thomson, 1999), following air masses from the position of the ship towards potential sources which are assumed to be in the lowest 500 m or 1000 m. For the VOC enhancements observed on 13 October, the

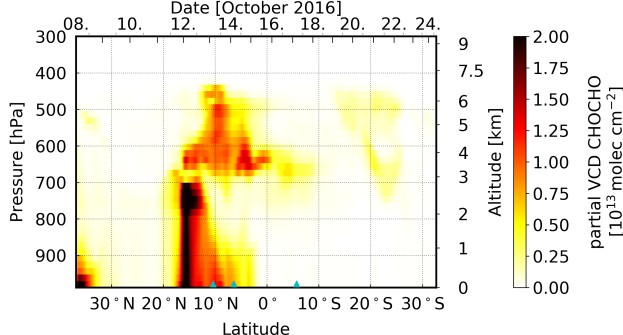

**Figure 14.** CHOCHO MOZART-4 profiles as used for VCDs calculations, interpolated on the cruise track (from North to South). The blue triangles on the bottom indicate the position of RV Maria S. Merian on the days with unusual scan angle dependency. For the calculation of the altitude, a mean temperature profile is used.

simulated emission sensitivities indicate air masses originating from the Sahel and the more southerly forest, shrub-, grass-, and croplands (Mayaux et al., 2004; Fig. 15) were measured at the vessel. These areas are potential source regions for emissions of biogenic precursors of HCHO and CHOCHO. Furthermore, fires from the FINN dataset are shown in Fig. 15. Some of these fires were detected in the possible source regions which can also be a source for VOCs. The simulated particles reach the measurement location over the open ocean at altitudes above 1500 m up to 3000 m after 2 days (Fig. 15) or longer (not shown), depending on the source region. Thus, the measured VOC enhancements are probably caused by export of precursor molecules from the African continent due to the short lifetime of HCHO and CHOCHO.

The results are different for the 14 October when the measurements also show enhanced HCHO and CHOCHO values. The emission sensitivity response function is generally low for the lowest 1000 m (Fig. 16; similar results were also found for the lowest 500 m, not shown). Only air older than 2 days (not shown) might originate from the continent, but also 4 days old air has only a small sensitivity to the continent (Fig. 16). The highest emission sensitivity to the lowest 1000 m is found above the open ocean south of the vessel's position. Furthermore, fires are unlikely to be important for the observed enhanced HCHO and CHOCHO values, as only a few small fires were detected in the potential continental source regions. Additionally, the source altitude is less clear due to similar emission sensitivities for different altitudes. The detected HCHO and CHOCHO might be located in an altitude between 1000 m to 5000 m.

For the HCHO enhancements observed on 17 October, the simulated emission sensitivities indicate air masses originating from southerly Central Africa (Angola, Democratic Republic of the Congo, Republic of the Congo, Gabon, and Republic of Equatorial Guinea; Fig. 17). The simulated air masses from the African continent reach the positions above the ship after 4 days in an altitude between 2000 m and 4500 m, consistent with MOZART-4 model simulations (Fig. 13). In the possible source regions, mostly different types of forests and grasslands were found (Mayaux et al., 2004) as well as fires with large emissions (Fig. 17). Both can emit precursors of HCHO. However, most fires are detected south of the possible source regions.

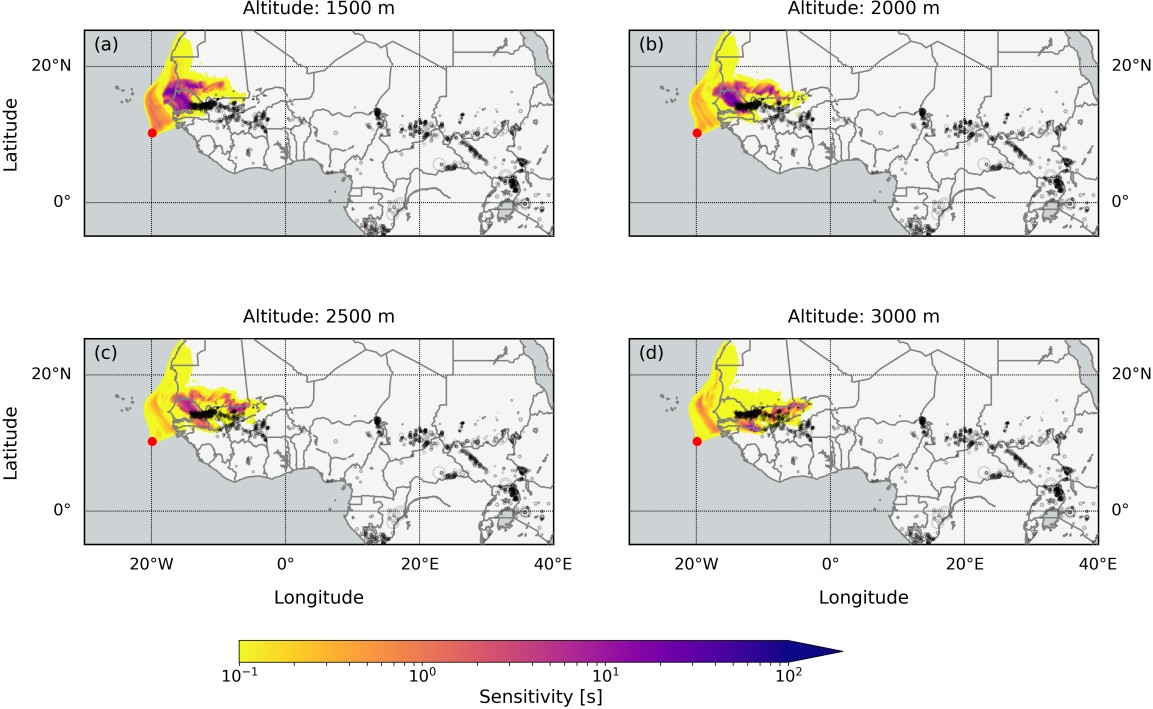

**Figure 15.** The emission sensitivity response function to the lowest 500 m layer for air parcels arriving at the receptor position in different altitudes above the RV Maria S. Merian on 13 October 2016 mid day, indicated by red dots (compare Fig. 9 and Fig. 11). The emission sensitivities were integrated over 2 days backward. The black circles are fires which were detected between 10 October and 12 October taken from the FINN database (Sect. 3.6). The circles are scaled with the calculated $CO_2$ emissions.

## 5 Comparison with previous studies

Several earlier studies showed that continental pollution can be transported over the open Atlantic Ocean, but none report a similar transport for VOCs. Anderson et al. (1996) found outflow from the African continent in the Southern Hemisphere. They analysed measurements from a flight campaign in September/October 1992 in the south Atlantic Ocean and found enhanced aerosol number densities at 3000 m to 4000 m with small loss of aerosol during the transport. However, their observations were mostly closer to the continent (distance between flight track and continent: $\sim 450 - 1500$ km, with one flight up to 3000 km) than the cruise track of MSM58/2, with large distances between the potential continental source and the area of the measurements (13 October 2016: $\sim 950$ km; 14 October 2016: $\sim 660$ km; between cruise track and continent, 17 October 2016: $\sim 2700$ km). Chatfield et al. (1998) showed export of CO in their model study, which was done for the same campaign.

While VOCs have not been reported over the Atlantic Ocean in connection to pollution outflow events until now, both HCHO and CHOCHO have been observed over the open ocean before. So far to our knowledge, there have been no reports on MAX-DOAS measurements of HCHO over the Atlantic Ocean. However, Weller et al. (2000) measured in-situ HCHO

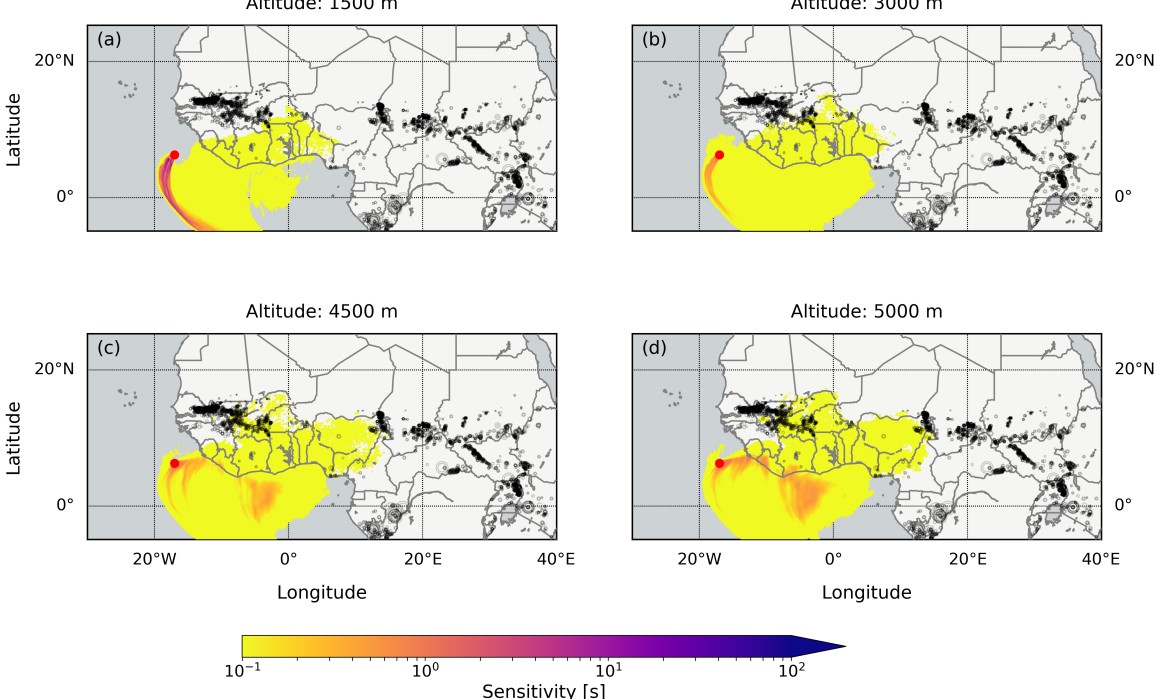

**Figure 16.** The emission sensitivity response function to the lowest 1000 m layer for air parcels arriving at the receptor position in different altitudes above the RV Maria S. Merian on 14 October 2016 mid day, indicated by red dots (compare Fig. 9 and Fig. 11). The emission sensitivities were integrated over 4 days backward. The black circles are fires which were detected between 8 October and 13 October taken from the FINN database (Sect. 3.6). The circles are scaled with the calculated $CO_2$ emissions. Only certain altitudes are shown as the altitudes 2000 m, 2500 m, 3500 m, and 4000 m have similar emission sensitivity pattern to the altitudes 1500 m and 3000 m.

concentrations in the Atlantic boundary layer during the Albatross campaign (October/November 1996). They found enhanced values in tropical latitudes, attributing them to secondary production in absence of any local sources, ruling out pollution transport due to the short lifetime of HCHO. Furthermore, Notholt et al. (2000) performed absorption measurements by Fourier transform infrared (FTIR) spectroscopy on the same cruise as Weller et al. (2000) to analyse i.a. HCHO. They found a similar latitudinal dependency as Weller et al. and converted their own measurements as well as the measurements performed by Weller et al. into total columns (assumed layer thickness: $0-9$ km). The retrieved columns are in the order of approximately $1.5 \times 10^{16}$ molec cm$^{-2}$ between $10°$ N and the Equator which agrees quite well with our MAX-DOAS measurements presented before. Also south of the Equator, they found values in the order of $1.0 \times 10^{16}$ molec cm$^{-2}$ with a further decrease towards the mid-latitudes. Generally, their results agree in magnitude and latitudinal distribution with our results. Nevertheless, there are slight differences compared to the previous findings, since the scan angle dependency of the enhanced HCHO dSCD on 13 October, 14 October, and 17 October clearly indicate an elevated layer of HCHO enhancements which was not shown in the two studies (most likely due to the different measurement techniques). Because of the good agreement between the

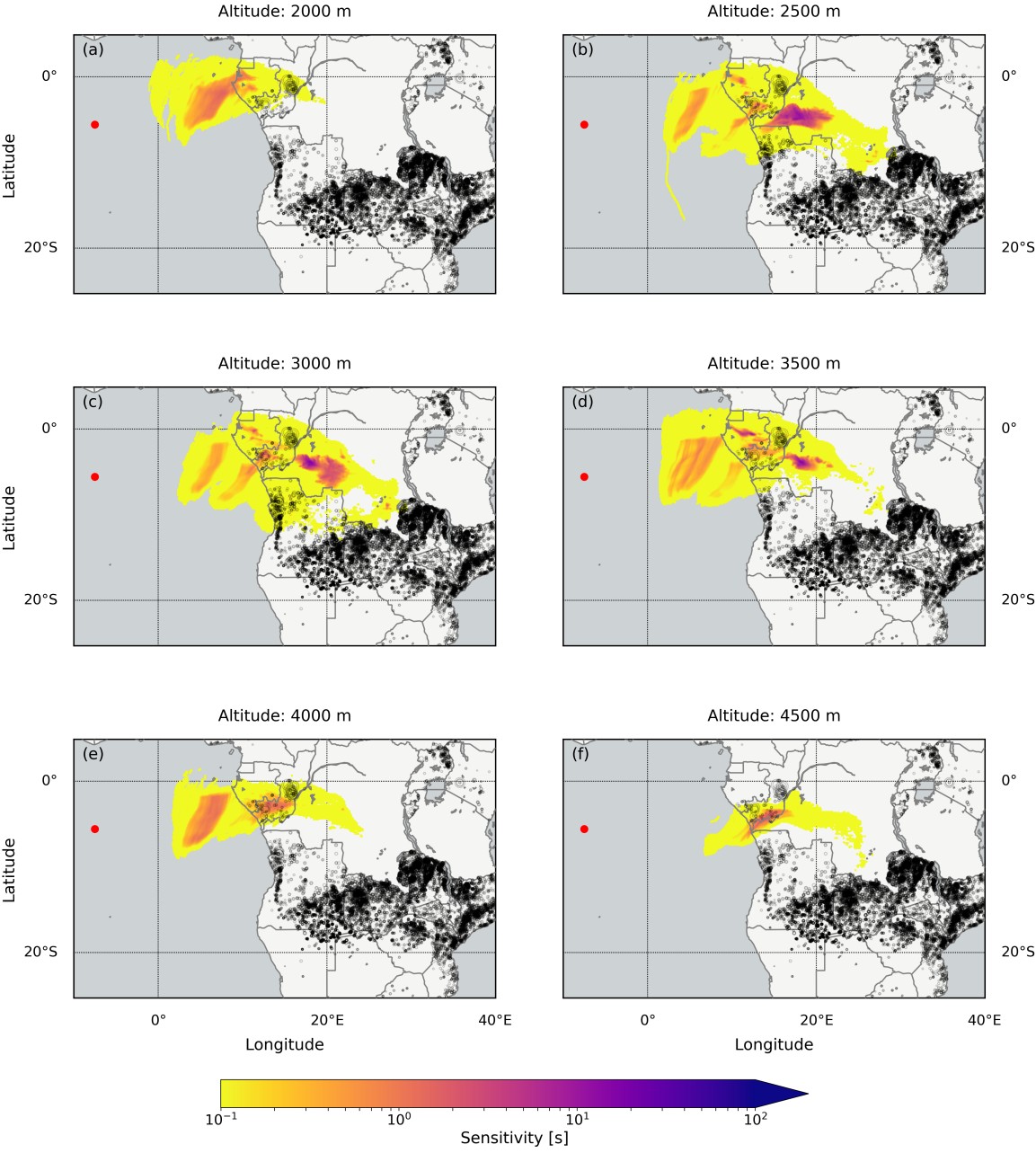

**Figure 17.** The emission sensitivity response function to the lowest 500 m layer for air parcels arriving at the receptor position in different altitudes above the RV Maria S. Merian on 17 October 2016 mid day, indicated by red dots (compare Fig. 9). The emission sensitivities were integrated over 4 days backward. The black circles are fires which were detected between 12 October and 14 October taken from the FINN database (Sect. 3.6). The circles are scaled with the calculated $CO_2$ emissions.

measurements of Notholt et al. and our measurements, this might indicate that the model data and the satellite measurements underestimate the amount of HCHO south of the Equator (Fig. 10).

Even though, in light of its short lifetime and missing sources over the open ocean, these enhanced levels of HCHO are surprising, several previous studies already have reported similar findings based on in satellite measurements (Wittrock et al., 2006, De Smedt et al., 2008, Meyer-Arnek et al., 2005). Meyer-Arnek et al. (2005), e.g., explained enhanced HCHO values over the tropical Atlantic Ocean by transport and transformation of VOC emissions. Furthermore, they could show that biomass burning and biogenic emissions produce similar amounts of HCHO, but that the transport of HCHO from biogenic emissions is in lower altitudes and usually stays closer to the source regions. Figures 15 – 17 show detected fires in October 2016 for Africa south of the Equator and also some fires for Africa north of the Equator in the potential source regions. On 13 October, fires are clearly visible in the potential source region. Thus, the results from Meyer-Arnek et al. (2005) support our present results, which were collected far away from the coast and indicate that the observed HCHO enhancements might be caused by biomass burning emissions of HCHO precursors. In contrast on 14 and 17 of October, the number of fires is limited, and therefore, biogenic origins might be more plausible sources for these regions. However, larger $NO_2$ columns from biomass burning were not detected, which could be related to the short lifetime of $NO_2$.

Also over the Pacific ocean enhanced HCHO columns were observed by Peters et al. (2012) and Tan et al. (2018). In both studies, HCHO was detected in elevated layer of 400 m and 500 m, which is lower than the detected altitudes in this study. Therefore, they concluded that the increased values are related to precursors of HCHO.

For CHOCHO, there have been MAX-DOAS measurements over the Atlantic during two cruises with RV Polarstern in 2009 and 2011 (Mahajan et al., 2014). These campaigns showed complex results: while in 2009 (when the cruise track was slightly more west compared to our observations here), no CHOCHO enhancements could be observed, the 2011 cruise, which followed a path very similar to the present study, showed clear CHOCHO dSCD enhancements of about $1 \times 10^{15}$ molec cm$^{-2}$ at around 5° N. In those data, no clear scan angle dependency could be identified, so the vertical location of the observed CHOCHO masses remains unclear. Our present results are consistent with these previous findings, showing slightly smaller CHOCHO dSCDs in the same area in an elevated layer. On the other hand, Sinreich et al. (2010) investigated CHOCHO over the Pacific Ocean, up to 3000 km away from continental land sources. They found strongly enhanced dSCDs of approximately $3 \times 10^{15}$ molec cm$^{-2}$ in the lowest elevation angles. Since their results show a clear scan angle dependency, Sinreich et al. concluded that CHOCHO was located in the marine boundary layer, pointing towards local production. On some days (for example 17 or 21 October 2016), the same scan angle dependency above the detection limit are found in our measurements during COPMAR, which suggest that besides CHOCHO in elevated layers also some CHOCHO was located close to the ground.

Similarly to HCHO, also CHOCHO has been reported to be present at enhanced levels over remote ocean regions in satellite observations (Vrekoussis et al., 2009; Stavrakou et al., 2009; Lerot et al., 2010). These CHOCHO enhancements are mostly visible in regions of strong biogenic activity and biomass burning, and are usually attributed to local production from CHOCHO precursors either originating from marine biota or from transported organic aerosol rich in dissolved organic carbon (Vrekoussis et al., 2009).

Our MAX-DOAS measurements suggest that the observed HCHO and CHOCHO enhancement is partly located in an elevated atmospheric layer which is in contrast to previous publications. It seems more likely that on the days where our measurements show enhanced values their source is related to transported precursors, which is also in line with the findings presented in Sect. 4.4. Possibly, the aerosols trap the gases which are lifted and transported together. These stored gases can then be re-released to the gas phase by reversible desorption after several days. Similar results were found for a case study over continental area in Asia by Alvarado (2016). The presence of a combination of dust and biogenic aerosol, and thus, potential VOC sources in that region and during that season could be shown by Ridley et al. (2012) using Cloud-Aerosol Lidar and Infrared Pathfinder Satellite Observations (CALIOP) and model data. Furthermore, Volkamer et al. (2015) found similar results for the Pacific Ocean. They found in the equatorial Pacific west of the American continent enhanced CHOCHO columns mostly in elevated layers, ruling out marine sources.

## 6 Summary and Conclusions

During the **C**ontinental **O**utflow of **P**ollutants towards the **MA**rine t**R**oposphere (COPMAR) project, a Multi-AXis Differential Absorption Spectrometer (MAX-DOAS) was operated on board of the research vessel (RV) Maria S. Merian for cruise MSM58/2, which was conducted from Ponta Delgada (Azores) to Cape Town (South Africa) in October 2016. The goal of the project was to investigate the enhanced quantities of formaldehyde (HCHO) and glyoxal (CHOCHO) frequently observed over the remote Atlantic Ocean in satellite measurements and model simulations.

We analysed our measurements for the latitudinal and diurnal variation of stratospheric $NO_2$, which are in good agreement with previous studies (e.g., Peters et al., 2012 and Kreher et al., 1995), showing proper operation of the instrument. This gives us confidence in the enhanced HCHO and CHOCHO columns observed in an elevated layer during the cruise in the area of expected outflow.

The MAX-DOAS observations of HCHO and CHOCHO show good or moderate agreement with satellite data and model simulations, with correlation coefficients of 0.72 (HCHO) and 0.55 (CHOCHO) between MAX-DOAS and MOZART-4 and between MAX-DOAS and satellite observations of 0.79 (OMI) and 0.72 (GOME-2B) for HCHO and 0.56 (OMI) for CHOCHO. MAX-DOAS HCHO were often higher than suggested by both satellite and model data. In contrast, CHOCHO MAX-DOAS observations are often lower than the model and satellite data. The enhanced HCHO levels which are observed in the Southern Hemisphere are not present in satellite and model data. This might be related to the fact that the latter two datasets are monthly means and the outflow event of an individual day cannot be resolved in these datasets.

For HCHO on 3 and for CHOCHO on 2 days, our measurements show clearly enhanced levels of these trace gases. The dependency of the measured dSCD on elevation angle suggests the presence of HCHO (on 13 October, 14 October and 17 October) and CHOCHO (on 13 October and 14 October only) in elevated atmospheric layers, most likely above the marine boundary layer. On 13 and 17 October, this clearly points to HCHO, CHOCHO, and/or their precursors being transported over long distances from the African continent. FLEXPART emission sensitivities for ship's position from potential source regions show the presence of air masses originating from the African continent in altitudes between 1500 m and 3000 as

well as between 2000 m and 4500 m in the Northern and Southern Hemisphere, respectively. This is in agreement with our observations. These air masses reach the measurement location over the Atlantic Ocean after 2 and 4 days. On 14 October, the results are less clear. Here, the emission sensitivity to the continent is small whereas the emission sensitivity is high to the open ocean. The air from the continent reaches the vessel's position after approximately 4 days in an altitude between 1000 m to 5000 m. MOZART-4 simulations show enhancements of both trace gases in an altitude between $3000 - 6000$ m and $3000 - 4000$ m in the Northern and Southern Hemisphere, respectively, which confirms our findings although the MOZART-4 altitudes are slightly higher in the Northern Hemisphere. These results are in general good agreement with previous studies of trace gases transported from Africa over the Atlantic Ocean.

Although our measurements do not show large levels of $NO_2$, these VOC enhancements probably originate from biomass burning on 13 October, as the source regions agree with fire detections from the FINN dataset. In contrast, for 17 October, only a small number of fires were observed in the potential source region, and therefore, biogenic origin might be the more realistic source. Thus, the main source of the detected VOC outflow is probably related to vegetation and/or biomass burning on the African continent.

The observed aerosol differs between Africa north and south of the Equator (having different AOD and Ångström-exponent) and therefore, the outflow in both hemispheres seems to be from different sources. While the aerosol observed in the Northern Hemisphere shows clear characteristics of desert dust, the aerosol measurements on 17 October (the day with enhanced HCHO values in elevated layers in the Southern Hemisphere) hint towards continental aerosol.

The present study is the first to confirm the enhanced levels of HCHO and CHOCHO frequently observed from satellites over the Atlantic Ocean using ship-based measurements. Our findings suggest that these enhanced levels of HCHO and CHOCHO are present in elevated atmospheric layers in the free troposphere, implying that these VOCs or their precursors are transported from the African continent or that we have re-released gases from a gas-aerosol combination. Further measurement campaigns should be conducted to investigate this pollution export in more detail, in order to shed light on the chemical transformations occurring in these plumes, and thus, enabling an explanation of the presence of short-lived species so far from their emission sources.

*Data availability.* The common cross sections used in this study are available from the cited references. The GOME-2 spectra (lv1b) and the OMI spectra (lv1) were provided by EUMETSAT and NASA (see also Acknowledgements) and are available from their respective websites. The lv2 and lv3 satellite data are available from the authors upon request. The MAX-DOAS data will be published on PANGAEA.

*Author contributions.* Lisa K. Behrens collected the data during the ship cruise MSM58/2, analysed the data, and combined the datasets. Andreas Hilboll, Andreas Richter, and Enno Peters provided additional feedback on the manuscript. Andreas Richter contributed the GOME-2A and GOME-2B $NO_2$ satellite data. Enno Peters designed the instrument and provided the program for the correction of the ship's movement. Leonardo M. A. Alvarado contributed the GOME-2B and OMI satellite data of HCHO and CHOCHO. Anna B. Kalisz Hedegaard and

Andreas Hilboll ran the FLEXPART simulations. Mihalis Vrekoussis conceived the COPMAR campaign. All co-authors contributed to the data interpretation and the manuscript preparation.

*Competing interests.*   The authors declare that they have no conflict of interest.

*Acknowledgements.*   This study has been funded by the University of Bremen, the state of Bremen, and by the DFG-Research Center / Cluster of Excellence "The Ocean in the Earth System". We further thank the "DFG Senatskommission für Ozeanographie" for the travelling time on RV Maria S. Merian and the "Leitstelle Deutsche Forschungsschiffe" for the logistical support. We acknowledge use of MOZART-4 global model output (available at http://www.acom.ucar.edu/wrf-chem/mozart.shtml). GOME-2 lv1b radiances have been provided by EUMETSAT and OMI lv1 data have been provided by NASA. Furthermore, we acknowledge the MAN effort of NASA's AERONET group led by Alexander Smirnov by providing a calibrated instrument and by maintaining the MAN data-base which is accessible via https://aeronet.gsfc.nasa.gov/new_web/maritime_aerosol_network.html. Anna Beata Kalisz Hedegaard's PhD work is financed through a DLR-DAAD Research Fellowship. All FLEXPART computations were performed at the "Aether" HPC cluster at the University of Bremen, funded by DFG within the scope of the Excellence Initiative.

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
