# Peer review of "Detection of Outflow of Formaldehyde and Glyoxal from the African continent to the Atlantic Ocean with a MAX-DOAS Instrument"

_Atmospheric Chemistry and Physics, 2018_

## Referee Comment (RC1) · Anonymous Referee #1 · 4 Feb 2019

This study presents first MAX-DOAS observations of formaldehyde and glyoxal in the Atlantic Ocean caused by transport of air masses coming from Africa. The paper is well written and the results appropriately discussed, with sufficient evidence to demonstrate that the observed signals of HCHO and CHOCHO are real and that their origin is relatively well understood. This is a valuable contribution to the understanding of VOC concentration levels that can be observed remotely from their known sources. A mechanism explaining how the HCHO/CHOCHO precursors could be transported so far away from their sources is proposed in the discussion, which would require further investigation but is beyond the scope of this work. This work is well suited for publication in ACP. I have only a few comments, which I ask to be considered before the

publication.

**Comments:**

- Page 15;line 29: The MAX-DOAS measurements do not show strongly enhanced values on 14th October, on contrary to MOZART-4 data. Please clarify and discuss this. It is also interesting to see that MOZART shows elevated values for that day for HCHO but not for glyoxal. Do you have any explanation for this?

- Page 16; lines 17-19: I don't find the explanation to rule out the AMF as potential source of the differences very convincing. If I understood well, profile shapes from MOZART are used for the slant to vertical column conversion in the MAX-DOAS analysis. Are those profiles also used for the satellite retrievals? Please mention which a priori profiles are used in the satellite retrievals. Because of the different observation geometries, an error in the a priori profile shape would impact differently the satellite and MAX-DOAS retrievals. Also, from Figs. 13 and 14, one can clearly see that the MOZART profile shapes change in time/latitude. So the argument that an error caused by the AMF would be constant in time doesn't seem valid to me. Could you add a figure showing the MAX-DOAS and satellite AMFs as a function of time/latitude so that we can better see what are their respective time/latitude variability's?

**Technical comments:**

- Page 9; line 12: remove either "several" or "different"

- Page 9; line 14-15: "In this study... satellite measurements." Phrase unclear. Please rephrase.

- Captions 4-5-6: replace "Example for" by "Example of"

- Eq. (7): replace "SDC_ref" by "SCD_ref"

- Page 11; line 6: replace "sensitivity for" by "sensitivity to"

- Page 11; line 19: replace "of degree of 5" by "of degree 5"

- Page 11; line 30: replace "lower SZA" by "larger SZA"

- Caption 7: replace "an linear" by "a linear"; add "than" in "smaller than 92°".

- Page 14; lines 2-3: remove "from Fig. 7" and "shown in Fig. 8"

- Page 14; line 5: add "the" in "related to the degradation"

- Figure 8 and following ones: I would suggest to mention in the caption that the latitudes are plotted from North to South, which is unusual.

- Page 18; line 10: Clarify why enhanced columns indicate that the satellite data are close to the detection limit. Having satellite columns higher than MAX-DOAS data might also be caused by non-zero glyoxal concentrations in the free troposphere (where the MAX-DOAS is not sensitive) or simply by artifacts in the satellite data. Could you add a sentence on this?

- Figures 13-14: I suggest adding altitude information along the y-axis to facilitate the link with the discussion.

- Caption figure 16: This figure refers to 14 October and not 13 I believe (2nd line)

- Caption figure 17: This figure refers to 17 October and not 13 I believe (2nd line)

- Page 23; line 6: Could you add information on the distance from the continent your cruise track was (more specifically for the days of the HCHO/glyoxal) hot spots.

- Page 23; line 32: For Fig 17, the number of fires seems rather limited in the source regions. For that particular case, the precursors are most likely from biogenic origin.

- Page 25; line 27: "these" instead of "this". Please make clear that the presented transport process in this paragraph is a potential explanation as there is no evidence presented in this study supporting particularly this.

- Page 23; line 18: "On 3 respectively 2 days" - Please rephrase.

- Page 23; line 33: See comment above - the number of fires in the source region is quite limited for 17th October.

---

## Referee Comment (RC2) · Anonymous Referee #2 · 12 Feb 2019

This manuscript describes the detection of HCHO and CHOCHO in African outflow during the COPMAR project in 2016. The authors find elevated HCHO and CHOCHO at higher altitudes, and suggest biomass burning and transport of long-lived precursor VOCs as the source. Overall, the manuscript presents an important set of observations and sufficient preliminary analysis and is suitable for ACP. The following comments should be addressed before publication:

Comments:

1.) Section 4.5 discusses MOZART outputs which show elevated HCHO and CHOCHO between 3000m and 6000m, while MAX-DOAS shows only elevated HCHO. FLEX-

PART is then used to investigate potential sources. Why not turn on/off biomass burning in MOZART and see the impact on modeled profiles, or look at MOZART outputs of other tracers?

2.) If MOZART underestimates HCHO south of the equator, how does using incorrect MOZART profiles to calculate MAX-DOAS VCDs influence the retrieval in those regions?

3.) I echo reviewer 1's request to discuss the profile shapes used in satellite retrievals. Ideally, there would be consistency between these profiles and those used in the MAX-DOAS retrievals.

4.) The last two paragraphs of section 5 (comparison with other studies) are important, and it is difficult to interpret the authors' intent.

The authors first state that CHOCHO enhancements are "usually attributed to local production from CHOCHO precursors either originating from marine biota or from transported organic aerosol rich in dissolved organic carbon". They then state that "it seems more likely that on this days [sic] its source is related to transported precursors". Are they referring to their own observations only, or commenting also on previous authors' conclusions?

The next sentences are also problematic. The authors write "The aerosols trap the gases which are lifted and transported together. These stored gases are then re-released to the gas phase by reversible desorption after several days." There is no evidence of this in their work. It should be stated as a potential explanation or hypothesis rather than fact.

5.) In general, there is not enough care given to instrument accuracy. A large portion of the manuscript is dedicated to describing the DOAS analysis. For that reason, uncertainty should be quantified if possible. AOD and Angstrom-exponent measurements are also discussed and compared without any comment on accuracy or precision.

[Figure]

Minor comments:

1.) Page 2, line 11: "A global background concentration of HCHO exists of 0.3 – 2.0 parts per billion". This is a wide range and it is unclear what is meant by 'background'. For this manuscript, a useful number would be the concentration (in ppb or column density) over the remote ocean stemming from CH4 oxidation.

2.) Section 2.4: What is meant by 'the model data are linearly interpolated?' Interpolated in time? Sampled at the observation location?

3.) Section 3.2: What versions of the satellite products are used? OMI-BIRA? OMI-SAO? The only reference given is a Ph.D. thesis. Either the product should be described, or peer reviewed literature should be sited.

4.) Section 3.6: Is FINN the inventory included in MOZART? If so, it should be in section 3.4.

5.) Page 9 lines 26-27: "For the calculation of this limit, different approaches were used in previous studies, for example, Peters et al. (2012), Sinreich et al. (2010), and Platt et al. (1997). In this study, the detection limit is calculated for each trace gas individually with the method from Sinreich et al. (2010)." If these methods are different, the choice should be justified. If they are not significantly different, the other studies should not be discussed.

6.) Page 15 line 17: "Due to the primary emission sources of HCHO" sounds like direct HCHO emissions, which I do not believe is what the authors mean. Perhaps "photochemical production of HCHO from VOC precursors".

7.) Page 27 line 1: Clarify what is meant by aerosol 'type'.

8.) Page 27 line 5: "The present study is the first to confirm the enhanced levels of HCHO and CHOCHO frequently observed from satellites over the Atlantic Ocean using ship-based measurements." Is this true (could be), or is it the first ship-based MAX-DOAS measurements over the Atlantic?

9.) Throughout: A correlation coefficient of 0.55 or 0.56 is 'moderate' (not 'good') agreement.

Technical comments:

1.) Page 9 line 14: "In this study, the trace gases are expected to be in elevated layers and from satellite measurements". Please clarify.

2.) Page 9 line 25: "ground-based" should be "ship-based".

3.) Page 26, line 18: It is unclear what is meant by "On 3 respectively 2 days".

---

## Author Comment (AC2) · 6 Jun 2019

**Author reply to Referee #2**

Lisa K. Behrens et al.

June 6, 2019

We thank Referee #2 for carefully reading our manuscript and for the helpful comments which will improve the quality of our manuscript. We will reply to the comments point by point.

Furthermore, we noticed a small mistake in the discussion manuscript. For the AMF calculation in the HCHO satellite retrieval, we accidentally used data from a different CTM (TM4-ECPL). However, for consistency, all retrievals in the manuscript should use the same CTM data as a-priori. Therefore, we have replaced the use of TM4-ECPL by the same MOZART-4 data as was also used in the MAX-DOAS and CHOCHO satellite retrievals. This introduces only small changes compared to the dataset shown in the discussion manuscript. In the revised manuscript, all HCHO and CHOCHO retrievals will use the same MOZART-4 data set.

Legend:
- referee comments
- authors comments
- text in manuscript
- **changed text in manuscript**

This manuscript describes the detection of HCHO and CHOCHO in African outflow during the COPMAR project in 2016. The authors find elevated HCHO and CHOCHO at higher altitudes, and suggest biomass burning and transport of long-lived precursor VOCs as the source. Overall, the manuscript presents an important set of observations and sufficient preliminary analysis and is suitable for ACP. The following comments should be addressed before publication:

Thank you very much for the positive comments.

**Comments:**

1.) Section 4.5 discusses MOZART outputs which show elevated HCHO and CHOCHO between 3000m and 6000m, while MAX-DOAS shows only elevated HCHO. FLEXPART is then used to investigate potential sources. Why not turn on/off biomass burning in MOZART and see the impact on modeled profiles, or look at MOZART outputs of other tracers?

Turning on/off biomass burning or other traces would be a good approach to investigate the differences between MOZART-4 and MAX-DOAS measurements. However, the MOZART-4 product which we use for the comparison and AMF calculation is an official product provided by NCAR through https://www.acom.ucar.edu/wrf-chem/mozart.shtml. Consequently, we did not run MOZART-4 by ourself and cannot run the model with different initial conditions. Furthermore, we suggested in the manuscript that the gases are trapped by aerosols and than transported. This mechanism cannot be shown by changes in the biomass burning initial conditions of MOZART-4.

In our study, we used FLEXPART simulations to gain further knowledge of the origin of the airmass which have been observed. FLEXPART can simulate the transport of such gases trapped by aerosols.

[Figure]

2.) If MOZART underestimates HCHO south of the equator, how does using incorrect MOZART profiles to calculate MAX-DOAS VCDs influence the retrieval in those regions?

Figure 1: HCHO MAX-DOAS BAMF for 30° VZA and 90° RAA for different SZAs.

The influence of the RAA on the AMF is often smaller than the influence of the SZA, therefore, in the following only the SZA will be discussed.

First, MOZART-4 can differ from the real atmosphere in several ways. As shown in Fig. 1, the influence on the AMF depends on the SZA. For small SZAs, BAMF variation with altitude is less pronounced than for large SZAs. Figure 13 in the paper shows that MOZART-4 simulates slightly enhanced HCHO values between 600 and 700 hPa on the days with large differences between the datasets. If this layer is in reality higher than 600 hPa or lower than 900 hPa and the SZA is small, the AMF would decrease and lead to higher VCDs of the MAX-DOAS measurements. Between 700 and 900 hPa, variability of the BAMF are small. Changes in these layers would result in a decrease of the MAX-DOAS VCDs. For large SZAs, the BAMF show a constant increase above 600 hPa leading again to a decrease of the MAX-DOAS VCDs, whereas for HCHO below 700 hPa, the MAX-DOAS values would be enhanced.

Second, the absolute amount of HCHO could be not correctly simulated. If the model underestimates/overestimates the amount of HCHO, this would in good approximation not change the AMF. Consequently, the MAX-DOAS VCDs would not change.

Third, the model could miss an additional layer of HCHO. This would also lead to an underestimation/overestimation of HCHO which results in larger VCDs for the MAX-DOAS measurements.

Because Referee#1 made a similar comment, we will change the discussion in Sect. 4.2 as follows:

[...] The MAX-DOAS measurements are mostly higher than the satellite observations (Fig. 10 a) which results in a slope larger than one (1.57, OMI and 1.10, GOME-2B) and a large offset ($0.86 \times 10^{15}$ molec cm$^{-2}$, OMI and $3.81 \times 10^{15}$ molec cm$^{-2}$, GOME-2B) of the regression line (Table 2). These differences are clearly visible between 20° N and 32° N (09 October – 11 October) and between 10° S and 22° S (18 October – 21 October; Fig. 10 a), which have been measured in clean remote ocean areas with low pollution.

**Compared to the MOZART-4 model values, the MAX-DOAS observations are often higher which can also be observed in a low slope of 0.68. Close to the Equator, the values show good agreement in the area of expected pollution outflow which leads to a high correlation coefficient between the two datasets of 0.72. However, the offset is also high between the two datasets ($4.82 \times 10^{15}$ molec cm$^{-2}$).**

**Several reasons can contribute to an enhancement in MAX-DOAS data. Large**

differences between the MAX-DOAS and the satellite or model data are found over regions with low air pollution north and south of the Equator. In these areas, high measurement uncertainties can be found in the satellite data due to the low columns which might influence the retrieved satellite values. It is also possible that the model and satellite results underestimate the VCDs, because of the potentially localised nature of the enhancements (see also Sect. 5). Increased uncertainties in the MAX-DOAS data in this region can be excluded as the DOAS fit RMS is nearly constant during the whole cruise (see Sect. 3.8.3). Furthermore, $H_2O$ interferences might contribute to the differences. Then, the differences should be reduced/increased for higher/lower elevation angles, respectively. However, this does not seem to be the case, as a similar behaviour is visible for all viewing directions. Additionally, bad weather conditions can be excluded, because the affected days of the MAX-DOAS measurements had different weather conditions (see Table 1) and an intensity filter was used to exclude poor viewing conditions (see Sect. 3.1). Similar conditions were also used for satellite values. Here, only measurements with geometric cloud fraction smaller 0.3 were included.

The AMF could introduce the differences between the datasets. However, this seems unlikely as the MOZART-4 model is used for the AMF calculations for both MAX-DOAS and satellite measurements and this model does not show a similar behaviour as the MAX-DOAS measurements. For example, if the model underestimates/overestimates the amount of HCHO in the atmosphere, this would in good approximation not change AMFs, and therefore, the VCDs of both datasets are not influenced. Thus, no difference would be introduced between the satellite and MAX-DOAS data. Possibly, the differences between the datasets can be related to the HCHO model profile. The HCHO profile of the model could differ from the real atmospheric HCHO profile. On the one hand, the model could miss an additional layer which is present in the atmosphere. Such a difference between model and real atmosphere would result in larger VCDs for the MAX-DOAS and satellite measurements. On the other hand, the model could simulate the HCHO in the wrong atmospheric layer. This could lead to both an underestimation or an overestimation of the MAX-DOAS VCDs depending on the profile and the SZA or relative azimuth angle which is similar for satellite measurements. Additionally, aerosols could influence the AMFs leading to differences in satellite and MAX-DOAS VCDs observations.

3.) I echo reviewer 1's request to discuss the profile shapes used in satellite retrievals. Ideally, there would be consistency between these profiles and those used in the MAX-DOAS retrievals.

We agree and therefore have used the same profiles for the AMF calculations for satellite and MAX-DOAS data for CHOCHO. In the revised version of the manuscript, we will also use MOZART-4 for the AMF calculation of HCHO for the satellite data. Unfortunately, we used TM4 in the manuscript for the AMF of satellite HCHO which should not have been the case. However, this change introduces only small changes for the comparison of the datasets. Please see also the answer to the previous question. We will add a comment on this in Sect. 3.4:

**The 4-D fields of HCHO and CHOCHO concentrations are needed as a priori information for the calculation of VCDs for both MAX-DOAS and satellite data (Sect. 3.7).**

4.) The last two paragraphs of section 5 (comparison with other studies) are important, and it is difficult to interpret the authors' intent.

The authors first state that CHOCHO enhancements are "usually attributed to local production from CHOCHO precursors either originating from marine biota or from transported organic aerosol rich in dissolved organic carbon". They then state that "it seems more likely that on this days [sic] its source is related to transported precursors". Are they referring to their own observations only, or commenting also on previous authors' conclusions?

The next sentences are also problematic. The authors write "The aerosols trap the gases which are lifted and transported together. These stored gases are then re-released to the gas phase by reversible desorption after several days." There is no evidence of this in their work. It should be stated as a potential explanation or hypothesis rather than fact.

We will change the text as follows:

[...] Similarly to HCHO, also CHOCHO has been reported to be present at enhanced levels over remote ocean regions in satellite observations (Vrekoussis et al., 2009; Stavrakou et al., 2009; Lerot et al., 2010). These CHOCHO enhancements are mostly visible in regions of strong biogenic activity and biomass burning, and are usually attributed to local production from CHOCHO precursors either originating from marine biota or from transported organic aerosol rich in dissolved organic carbon (Vrekoussis et al., 2009).

**Our MAX-DOAS measurements suggest that the observed** HCHO **and** CHOCHO **enhancement is partly located in an elevated atmospheric layer which is in contrast to previous publications. It seems more likely that on the days where our measurements show enhanced values its source is related to transported precursors, which is also in line with the findings presented in Sect. 4.4. Possibly, the aerosols trap the gases which are lifted and transported together. These stored gases can then be re-released to the gas phase by reversible desorption after several days.** Similar results were found for a case study over continental area in Asia by Alvarado (2016). The presence of a combination of dust and biogenic aerosol, and thus, potential VOC sources in that region and during that season could be shown by Ridley et al. (2012) using Cloud-Aerosol Lidar and Infrared Pathfinder Satellite Observations (CALIOP) and model data. Furthermore, Volkamer et al. (2015) found similar results for the Pacific Ocean. They found in the equatorial Pacific west of the American continent enhanced CHOCHO columns mostly in elevated layers, ruling out marine sources. [...]

5.) In general, there is not enough care given to instrument accuracy. A large portion of the manuscript is dedicated to describing the DOAS analysis. For that reason, uncertainty should be quantified if possible. AOD and Angstrom-exponent measurements are also discussed and compared without any comment on accuracy or precision.

For satellite measurements error analyses are discussed in the publications from Boersma et al. (2004) and Lorente et al. (2017) for $NO_2$ and HCHO. Furthermore, for the satellite products which we are using here the errors will be discussed in Alvarado et al. (2014, 2019). We will add the following sentences in the manuscript in Sect. 3.2:

**The measurement accuracy for satellite data are described in Boersma et al. (2004) and Lorente et al. (2017) as well as in Alvarado et al. (2014, 2019). Generally, over remote ocean areas the uncertainty of tropospheric columns is dominated by errors in the stratospheric column (for** $NO_2$**) and errors in the spectral fitting (Boersma et al., 2004). Boersma et al. (2004) found a relative uncertainty of up to 100 % for** $NO_2$ **in these areas.**

For the AOD and Ångström-exponent we used publicly available products which can be downloaded at: https://aeronet.gsfc.nasa.gov/new_web/maritime_aerosol_network.html. Details about

this product can be found in Smirnov et al. (2009). Measurements errors for this instrument are discussed in Ichoku et al. (2002), Morys et al. (2001), Porter et al. (2001) and Knobelspiesse et al. (2003, 2004). We will add the following sentence in the manuscript in Sect. 3.3:

**Details of the Microtops handheld Sun photometers and their uncertainties can be found in Ichoku et al. (2002), Morys et al. (2001), Porter et al. (2001) and Knobelspiesse et al. (2003, 2004).**

MAX-DOAS measurements are influence by systematic and random errors. The following paragraph will be added in Sect. 3.8.1:

**Each MAX-DOAS measurement is influenced by a systematic and a random error (Irie et al., 2011; Takashima et al., 2012). The random error can be estimated from the residual of the fitted trace gases (Takashima et al., 2012). However, also systematic errors can be found in the residual such as not perfectly resolved cross sections. Furthermore, Boersma et al. (2004) and Takashima et al. (2012) showed that the temperature dependency of the cross section also leads to systematic errors in the retrieved SCD. For example for $NO_2$, it is in the order of $0.4\,\%\,K^{-1}$ (Takashima et al., 2012). For the vertical columns, the errors are in the order of 15 to $20\,\%$ (Takashima et al., 2012; Peters et al., 2012).**

**Minor comments:**

1.) Page 2, line 11: "A global background concentration of HCHO exists of 0.3 – 2.0 parts per billion". This is a wide range and it is unclear what is meant by 'background'. For this manuscript, a useful number would be the concentration (in ppb or column density) over the remote ocean stemming from CH4 oxidation.

We will change the text as follows:

[...] Primary emissions are from biomass burning and the combustion of fossil fuels, but HCHO is also formed in the atmosphere from the oxidation of methane ($CH_4$) and non-methane hydrocarbons (Arlander et al., 1995). **Due to the oxidation of $CH_4$, HCHO is not only found close to its source regions, but a global background concentration of HCHO exists with surface levels of $0.2-1.0$ parts per billion in remote marine environments (ppb; Weller et al., 2000; Burkert et al., 2001; Singh et al., 2001).** Furthermore, it is an important indicator of the photochemical activity for a region (De Smedt et al., 2008; Vrekoussis et al., 2010). [...]

2.) Section 2.4: What is meant by 'the model data are linearly interpolated?' Interpolated in time? Sampled at the observation location?

Yes, in Section 3.4 a interpolation in time is meant. We will change the text as follows:

[...]**For the comparison, the model data are linearly interpolated in time on the cruise track.**

3.) Section 3.2: What versions of the satellite products are used? OMI-BIRA? OMI-SAO? The

only reference given is a Ph.D. thesis. Either the product should be described, or peer reviewed literature should be sited.

The satellite retrievals used in this study are in house products from IUP Bremen based on OMI level 1 data provided by NASA (Levelt et al., 2006) and GOME2 level 1 data from EUMETSAT (Callies et al., 2000; Munro et al., 2016). For NO$_2$, we used the fit settings presented in Richter et al. (2011). For HCHO and CHOCHO, the settings were presented in Alvarado et al. (2018) at EGU which will be published in Alvarado et al. (2019). The CHOCHO is an improved version of Alvarado et al. (2014). We will additionally include in the manuscript the reference Alvarado et al. (2018):

[...] **Additionally, satellite observations for** HCHO **and** CHOCHO **are taken into account which are described in Alvarado et al. (2018) and Alvarado et al. (2019).** [...]

4.) Section 3.6: Is FINN the inventory included in MOZART? If so, it should be in section 3.4.

No, MOZART uses the GFED-v2 inventory. We use FINN dataset in Sect. 4.5 to discuss potential sources of VOC precursors.

5.) Page 9 lines 26-27: "For the calculation of this limit, different approaches were used in previous studies, for example, Peters et al. (2012), Sinreich et al. (2010), and Platt et al. (1997). In this study, the detection limit is calculated for each trace gas individually with the method from Sinreich et al. (2010)." If these methods are different, the choice should be justified. If they are not significantly different, the other studies should not be discussed.

As suggested by the referee, we will remove the sentence: "For the calculation of this limit, different approaches were used in previous studies, for example, Peters et al. (2012), Sinreich et al. (2010), and Platt et al. (1997)." from the manuscript.

6.) Page 15 line 17: "Due to the primary emission sources of HCHO" sounds like direct HCHO emissions, which I do not believe is what the authors mean. Perhaps "photochemical production of HCHO from VOC precursors".

No, we meant primary emissions. The formulation might be misleading and we will change the manuscript as follows:

The latitudinal variation of daily mean MAX-DOAS HCHO dSCDs observed during the cruise shows enhanced values in an elevated layer at both $\sim 10°\,$N and $\sim 5°\,$S. This coincides with the area of expected outflow from the African continent as seen in satellite measurements (Fig. 1 a). **Typically, the** HCHO **concentration in polluted areas is expected to be highest close to the surface (De Smedt et al., 2008; Heckel et al., 2005) due to the primary emission sources of** HCHO **and rapid photochemical production from VOC precursors. Thus, also the highest dSCDs should be observed at low elevation angles near the sources.** Figure 9 illustrates the daily mean HCHO dSCDs during the cruise at different elevation angles. [...]

7.) Page 27 line 1: Clarify what is meant by aerosol 'type'.

We will change the sentence as follows:

[...] **The observed aerosol differs between Africa north and south of the Equator (having different AOD and Ångström-exponent) and therefore, the outflow in both hemispheres seems to be from different sources.** [...]

8.) Page 27 line 5: "The present study is the first to confirm the enhanced levels of HCHO and CHOCHO frequently observed from satellites over the Atlantic Ocean using ship-based measurements." Is this true (could be), or is it the first ship-based MAX-DOAS measurements over the Atlantic?

This were not the first MAX-DOAS measurements over the Atlantic Ocean, but the first where both, HCHO and CHOCHO, were analysed and connected to outflow from the African continent. Mahajan et al. (2014) analysed CHOCHO MAX-DOAS measurements over the Atlantic Ocean. To our knowledge, HCHO has previously only been analysed from other measurements types.

9.) Throughout: A correlation coefficient of 0.55 or 0.56 is 'moderate' (not 'good') agreement.

Changed as suggested.

**Technical comments:**

1.) Page 9 line 14: "In this study, the trace gases are expected to be in elevated layers and from satellite measurements". Please clarify.

"In this study, the trace gases are expected to be in elevated layers and from satellite measurements. Furthermore, it is expected that they have a latitudinal dependency (see Fig. 1)."
We will change the manuscript as follows:

[...]**In this study, the trace gases are expected to be in elevated layers. Furthermore, satellite measurements indicate that they depend on latitude (see Fig. 1).**[...]

2.) Page 9 line 25: "ground-based" should be "ship-based".

Changed as suggested.

3.) Page 26, line 18: It is unclear what is meant by "On 3 respectively 2 days".

We will change the text as follows:

[...]**For HCHO on 3 and for CHOCHO on 2 days, our measurements show clearly enhanced levels of these trace gases.** [...]

Additionally as suggested by Referee #1, we will change three main points in the revised manuscript:

1. We clarified the discussion of HCHO for the days 13, 14, and 17 October.

2. We added an altitude axis to Fig. 13 and Fig. 14.

3. We added the distances to the continent/hot spot area for the days with observed outflow events.

**References**

[revised manuscript text omitted]

---

## Author Response (AR1)

**Revised manuscript**

Dear Dr. Steven Brown,

we would like to take the opportunity to thank you for your efforts and that you accepted the editorship of our manuscript "Detection of Outflow of Formaldehyde and Glyoxal from the African continent to the Atlantic Ocean with a MAX-DOAS Instrument". Furthermore, we would like to thank you for the extension of the review period for the manuscript.

Please find enclosed a revised version of our manuscript where we implemented all comments by the referees. The original manuscript has been revised according to their suggestions. We clarified all parts that where not formulated clear enough, in particular the differences between the model data and our measurements. Furthermore, we discussed the influence of the model data on the conversion from SCD to VCD.

We noticed a small mistake in the discussion manuscript. For the AMF calculation in the HCHO satellite retrieval, we accidentally used data from a different CTM (TM4-ECPL). In the revised manuscript, we have replaced the use of TM4-ECPL by the same MOZART-4 data. Now, all HCHO and CHOCHO retrievals will use the same MOZART-4 data set. This introduces only small changes compared to the dataset shown in the discussion manuscript. In addition to these changes and changes motivated by the reviewers, we included further information in Section 3.5 about details for the FLEXPART simulations.

Below the author comments are provided that have already been uploaded to the ACP web page on 6 June 2019. We also provide here a version of the revised manuscript in which changes in comparison to the initial version are highlighted. We hope that with the submission of the author comments and the revision of the manuscript, our article will be accepted for publication in ACP.

Yours sincerely, Lisa Behrens (on behalf of the co-authors)

List of Attachments

- Author comments to Referee #1
- Author comments to Referee #2
- Revised manuscript with highlighted changes
- Revised manuscript

**Author reply to Referee #1**

**Lisa K. Behrens et al.**

**June 6, 2019**

We thank Referee #1 for carefully reading our manuscript and for the helpful comments which will improve the quality of our manuscript. We will reply to the comments point by point.

Furthermore, we noticed a small mistake in the discussion manuscript. For the AMF calculation in the HCHO satellite retrieval, we accidentally used data from a different CTM (TM4-ECPL). However, for consistency, all retrievals in the manuscript should use the same CTM data as a-priori. Therefore, we have replaced the use of TM4-ECPL by the same MOZART-4 data as was also used in the MAX-DOAS and CHOCHO satellite retrievals. This introduces only small changes compared to the dataset shown in the discussion manuscript. In the revised manuscript, all HCHO and CHOCHO retrievals will use the same MOZART-4 data set.

Legend:

- referee comments
- authors comments
- text in manuscript
- changed text in manuscript

This study presents first MAX-DOAS observations of formaldehyde and glyoxal in the Atlantic Ocean caused by transport of air masses coming from Africa. The paper is well written and the results appropriately discussed, with sufficient evidence to demonstrate that the observed signals of HCHO and CHOCHO are real and that their origin is relatively well understood. This is a valuable contribution to the understanding of VOC concentration levels that can be observed remotely from their known sources. A mechanism explaining how the HCHO/CHOCHO precursors could be transported so far away from their sources is proposed in the discussion, which would require further investigation but is beyond the scope of this work. This work is well suited for publication in ACP. I have only a few comments, which I ask to be considered before the publication.

Thank you very much for the positive comments.

**Comments:**

Page 15; line 29: The MAX-DOAS measurements do not show strongly enhanced values on 14th October, on contrary to MOZART-4 data. Please clarify and discuss this. It is also interesting to see that MOZART shows elevated values for that day for HCHO but not for glyoxal. Do you have any explanation for this?

We clarified in the discussion that the MAX-DOAS measurements are not strongly enhanced on 14th October, in contrast to MOZART-4 data. We agree with the referee that the difference between HCHO and CHOCHO is interesting, but unfortunately we have no explanation for this difference. We will change the manuscript as follows:

[...] When converting the measured HCHO dSCD to VCDs (using the  $30^{\circ}$  elevation angle and HCHO profiles from the MOZART-4 model), the values on 13 October, 14 October, and 17 October remain clearly enhanced (see Fig. 10 a). At least the first region sampled on 13/14 October (~10° N) indeed coincides with the area for which satellite observations regularly show enhanced HCHO values (compare Fig. 1 a and Fig. 2). On these days, also MOZART-4 data are enhanced in the region. However, differences between the datasets are visible. On 13 October, MAX-DOAS and MOZART-4 data are higher than the satellite data. While on 14 October, MAX-DOAS and satellite data are slightly smaller than on 13 October, MOZART-4 data are further increased. On 17 October ( $5^{\circ}$  S), there is no clear sign of a HCHO enhancement in the satellite and model data in contrast to MAX-DOAS measurements. These differences could be related to several reasons. The MAX-DOAS data are measured on single days, whereas the satellite datasets are monthly means and MOZART-4 time series are interpolated on the cruise track. Thus, the differences between MAX-DOAS and satellite data could be explained by single, isolated outflow events on that particular days, which are not distinguishable from background values in the monthly averaged satellite data. The interpolation of the model data could lead to differences between model and MAX-DOAS data as for example isolated events might not be represented in the model data. Furthermore, the satellite and model data are averages over larger areas which could also influence the magnitude of the measured or simulated values. Thus, comparing HCHO columns retrieved from OMI and GOME-2B radiances (Sect. 3.2) and integrated columns from simulated MOZART-4 profiles (Sect. 3.4; interpolated on the cruise track) to our MAX-DOAS VCDs, the results generally confirm the finding of enhanced HCHO satellite and model columns. The datasets show good agreements with correlation coefficients larger than 0.70 (Table 2). [...]

Page 16; lines 17-19: I don't find the explanation to rule out the AMF as potential source of the differences very convincing. If I understood well, profile shapes from MOZART are used for the slant to vertical column conversion in the MAX-DOAS analysis. Are those profiles also used for the satellite retrievals? Please mention which a priori profiles are used in the satellite retrievals. Because of the different observation geometries, an error in the a priori profile shape would impact differently the satellite and MAX-DOAS retrievals. Also, from Figs. 13 and 14, one can clearly see that the MOZART profile shapes change in time/latitude. So the argument that an error caused by the AMF would be constant in time doesn't seem valid to me. Could you add a figure showing the MAX-DOAS and satellite AMFs as a function of time/latitude so that we can better see what are their respective time/latitude variability's?

We agree and therefore have used the same profiles for the AMF calculations for satellite and MAX-DOAS data for CHOCHO. In the revised version of the manuscript, we will also use MOZART-4 for the AMF calculation of HCHO for the satellite data. Unfortunately, we used TM4 in the manuscript for the AMF of satellite HCHO which should not have been the case. However, this change introduces only small changes for the comparison of the datasets. We will add a comment on this in Sect. 3.4:

The 4-D fields of HCHO and CHOCHO concentrations are needed as a priori information for the calculation of VCDs for both MAX-DOAS and satellite data (Sect. 3.7).

Figure 1: Daily mean HCHO AMF for ship-based (30° VZA) and satellite data.

Figure 1 shows a time series of the AMFs which show only small changes in the AMFs of MAX-DOAS and satellite data especially for the days of interest. A similar comment was made by Referee#2. Thus, we will change the discussion in Sect. 4.2 as follows:

[...] The MAX-DOAS measurements are mostly higher than the satellite observations (Fig. 10 a) which results in a slope larger than one (1.57, OMI and 1.10, GOME-2B) and a large offset  $(0.86 \times 10^{15} \text{ molec cm}^{-2}, \text{OMI} \text{ and } 3.81 \times 10^{15} \text{ molec cm}^{-2}, \text{ GOME-2B})$  of the regression line (Table 2). These differences are clearly visible between 20° N and 32° N (09 October – 11 October) and between 10° S and 22° S (18 October – 21 October; Fig. 10 a), which have been measured in clean remote ocean areas with low pollution.

Compared to the MOZART-4 model values, the MAX-DOAS observations are often higher which can also be observed in a low slope of 0.68. Close to the Equator, the values show good agreement in the area of expected pollution outflow which leads to a high correlation coefficient between the two datasets of 0.72. However, the offset is also high between the two datasets  $(4.82 \times 10^{15} \text{ molec cm}^{-2})$ .

Several reasons can contribute to an enhancement in MAX-DOAS data. Large differences between the MAX-DOAS and the satellite or model data are found over regions with low air pollution north and south of the Equator. In these areas, high measurement uncertainties can be found in the satellite data due to the low columns which might influence the retrieved satellite values. It is also possible that the model and satellite results underestimate the VCDs, because of the potentially localised nature of the enhancements (see also Sect. 5). Increased uncertainties in the MAX-DOAS data in this region can be excluded as the DOAS fit RMS is nearly constant during the whole cruise (see Sect. 3.8.3). Furthermore,  $H_2O$  interferences might contribute to the differences. Then, the differences should be reduced/increased for higher/lower elevation angles, respectively. However, this does not seem to be the case, as a similar behaviour is visible for all viewing directions. Additionally, bad weather conditions can be excluded, because the affected days of the MAX-DOAS measurements had different weather conditions (see Table 1) and an intensity filter was used to exclude poor viewing conditions (see Sect. 3.1). Similar conditions were also used for satellite values. Here, only measurements with geometric cloud fraction smaller 0.3 were included.

The AMF could introduce the differences between the datasets. However, this seems unlikely as the MOZART-4 model is used for the AMF calculations for both MAX-DOAS and satellite measurements and this model does not show a similar behaviour as the MAX-DOAS measurements. For example, if the model underestimates/overestimates the amount of HCHO in the atmosphere, this would in good approximation not change AMFs, and therefore, the VCDs of both datasets are not influenced. Thus, no difference would be introduced between the satellite and MAX-DOAS data. Possibly, the differences between the datasets can be related to the HCHO model profile. The HCHO profile of the model could differ from the real atmospheric HCHO profile. On the one hand, the model could miss an additional layer which is present in the atmosphere. Such a difference between model and real atmosphere would result in larger VCDs for the MAX-DOAS and satellite measurements. On the other hand, the model could simulate the HCHO in the wrong atmospheric layer. This could lead to both an underestimation or an overestimation of the MAX-DOAS VCDs depending on the profile and the SZA or relative azimuth angle which is similar for satellite measurements. Additionally, aerosols could influence the AMFs leading to differences in satellite and MAX-DOAS VCDs observations.

**Technical comments:**

Page 9; line 12: remove either "several" or "different".

"several" has been removed.

Page 9; line 14-15: "In this study... satellite measurements." Phrase unclear. Please rephrase.

"In this study, the trace gases are expected to be in elevated layers and from satellite measurements. Furthermore, it is expected that they have a latitudinal dependency (see Fig. 1)." We will change the manuscript as follows:

[...]In this study, the trace gases are expected to be in elevated layers. Furthermore, satellite measurements indicate that they depend on latitude (see Fig. 1).[...]

Captions 4-5-6: replace "Example for" by "Example of"

Done.

Eq. (7): replace "SDC\_ref" by "SCD\_ref" Done.

Page 11; line 6: replace "sensitivity for" by "sensitivity to" Done.

Page 11; line 19: replace "of degree of 5" by "of degree 5" Done.

Page 11; line 30: replace "lower SZA" by "larger SZA"

Done.

Caption 7: replace "an linear" by "a linear"; add "than" in "smaller than 92°".

Done.

Page 14; lines 2-3: remove "from Fig. 7" and "shown in Fig. 8"

Done.

Page 14; line 5: add "the" in "related to the degradation"

Done.

Figure 8 and following ones: I would suggest to mention in the caption that the latitudes are plotted from North to South, which is unusual.

Changed as suggested.

Page 18; line 10: Clarify why enhanced columns indicate that the satellite data are close to the detection limit. Having satellite columns higher than MAX-DOAS data might also be caused by non-zero glyoxal concentrations in the free troposphere (where the MAX-DOAS is not sensitive) or simply by artifacts in the satellite data. Could you add a sentence on this?

**We agree with the referee that the formulation might be misleading and clarified this paragraph. We will change the sentence as follows:**

[...] CHOCHO VCDs from all three datasets show enhanced values around 10° N while MAX-DOAS and MOZART-4 values further north and further south are close to zero. In comparison, OMI observations show enhanced CHOCHO columns throughout the tropics in Fig. 12. This behaviour could be explained by an elevated CHOCHO layer which is not represented in the model and cannot be detected by MAX-DOAS measurements. The enhanced values throughout the tropics are also represented in the slope of the regression line with 0.32, nevertheless only a negligible offset of  $-0.05 \times 10^{15}$  molec cm-2 was found (Table 3). The correlation between MAX-DOAS and OMI CHOCHO VCDs is 0.56. [...]

**Figures 13-14: I suggest adding altitude information along the y-axis to facilitate the link with the discussion.**

Changed as suggested. We will add an approximate corresponding altitude along the y-axis which is retrieved from a mean temperature profile retrieved from the model data. The mean temperature profile is used, because of the changing temperature and pressure conditions during the cruise. The results are linked to the discussion.

Figure 2: HCHO MOZART-4 profiles as used for VCDs calculations, interpolated on the cruise track (from North to South). The blue triangles on the bottom indicate the position of RV Maria S. Merian on the days with unusual scan angle dependency. For the calculation of the altitude, a mean temperature profile is used.

Figure 3: CHOCHO MOZART-4 profiles as used for VCDs calculations, interpolated on the cruise track (from North to South). The blue triangles on the bottom indicate the position of RV Maria S. Merian on the days with unusual scan angle dependency. For the calculation of the altitude, a mean temperature profile is used.

Caption figure 16: This figure refers to 14 October and not 13 I believe (2nd line)

Yes. We changed the caption.

Caption figure 17: This figure refers to 17 October and not 13 I believe (2nd line)

Yes. We changed the caption.

Page 23; line 6: Could you add information on the distance from the continent your cruise track was (more specifically for the days of the HCHO/glyoxal) hot spots.

Distance from the hot spot on 13 October 2016 to the cruise track:  $\sim 950$  km. Distance from the coast of the continent on 14 October 2016 to the cruise track:  $\sim 660$  km. Distance from the hot spot on 17 October 2016 to the cruise track:  $\sim 2700$  km. We will change the text as follows:

Several earlier studies showed that continental pollution can be transported over the open Atlantic Ocean, but none report a similar transport for VOCs. Anderson et al. (1996) found outflow from the African continent in the Southern Hemisphere. They analysed measurements from a flight campaign in September/October 1992 in the south Atlantic Ocean and found enhanced aerosol number densities at 3000 m to 4000 m with small loss of aerosol during the transport. However, their observations were partly closer to the continent (distance between flight track and continent: mostly  $\sim 450-1500$  km, one flight up to 3000 km) than the cruise track of MSM58/2 with large distances between the potential continental source and the area of the measurements (13 October 2016:  $\sim 950$  km; 14 October 2016:  $\sim 660$  km; between cruise track and continent, 17 October 2016:  $\sim 2700$  km). Chatfield et al. (1998) showed export of CO in their model study, which was done for the same campaign. [...]

Page 23; line 32: For Fig 17, the number of fires seems rather limited in the source regions. For that particular case, the precursors are most likely from biogenic origin.

**That is true, but there have been some large fires during that time. Nevertheless, biogenic origin might the more plausible source. Therefore, we will be more specific in the discussion. We will change the text as follows:**

[...] Figures 15-17 show detected fires in October 2016 for Africa south of the Equator and also some fires for Africa north of the Equator in the potential source regions. On 13 October, fires are clearly visible in the potential source region. Thus, the results from Meyer-Arnek et al. (2005) support our present results, which were collected far away from the coast and indicate that the observed HCHO enhancements might be caused by biomass burning emissions of HCHO precursors. In contrast on 14 and 17 of October, the number of fires is limited, and therefore, biogenic origins might be more plausible sources for these regions. However, larger NO2 columns from biomass burning were not detected, which could be related to the short lifetime of NO2. [...]

Page 25; line 27: "these" instead of "this". Please make clear that the presented transport process in this paragraph is a potential explanation as there is no evidence presented in this study supporting particularly this.

**We will change the text as follows:**

[...] Similarly to HCHO, also CHOCHO has been reported to be present at enhanced levels over remote ocean regions in satellite observations (Vrekoussis et al., 2009; Stavrakou et al., 2009; Lerot et al., 2010). These CHOCHO enhancements are mostly visible in regions of strong biogenic activity and biomass burning, and are usually attributed to local production from CHOCHO precursors either originating from marine biota or from transported organic aerosol rich in dissolved organic carbon (Vrekoussis et al., 2009).

Our MAX-DOAS measurements suggest that the observed HCHO and CHOCHO enhancement is partly located in an elevated atmospheric layer which is in contrast to previous publications. It seems more likely that on the days where our measurements show enhanced values its source is related to transported precursors, which is also in line with the findings presented in Sect. 4.4. Possibly, the aerosols trap the gases which are lifted and transported together. These stored gases can then be re-released to the gas phase by reversible desorption after several days. Similar results were found for a case study over continental area in Asia by Alvarado (2016). The presence of a combination of dust and biogenic aerosol, and thus, potential VOC sources in that region and during that season could be shown by Ridley et al. (2012) using Cloud-Aerosol Lidar and Infrared Pathfinder Satellite Observations (CALIOP) and model data. Furthermore, Volkamer et al. (2015) found similar results for the Pacific Ocean. They found in the equatorial Pacific west of the American continent enhanced CHOCHO columns mostly in elevated layers, ruling out marine sources. [...]

**Page 23; line 18: "On 3 respectively 2 days" - Please rephrase.**

We will change the text as follows:

[...]For HCHO on 3 and for CHOCHO on 2 days, our measurements show clearly enhanced levels of these trace gases.[...]

Page 23; line 33: See comment above - the number of fires in the source region is quite limited for 17th October.

**We will change the text as follows:**

[...] Although our measurements do not show large levels of NO2, these VOC enhancements probably originate from biomass burning on 13 October, as the source regions agree with fire detections from the FINN dataset. In contrast, for 17 October, only a small number of fires were observed in the potential source region, and therefore, biogenic origin might be the more realistic source. Thus, the main source of the detected VOC outflow is probably related to vegetation and/or biomass burning on the African continent.

The observed aerosol differs between Africa north and south of the Equator (having different AOD and Ångström-exponent) and therefore, the outflow in both hemispheres seems to be from different sources. [...]

Additionally as suggested by Referee #2, we will change two main points in the revised manuscript:

- 1. We clarified the discussion about transported precursors.
- 2. We included details about uncertainties.

**References**

- Alvarado, L. M. A.: Investigating the role of glyoxal using satellite and MAX-DOAS measurements, Ph.D. thesis, University of Bremen, 2016.
- Anderson, B. E., Grant, W. B., Gregory, G. L., Browell, E. V., Collins, J. E., Sachse, G. W., Bagwell, D. R., Hudgins, C. H., Blake, D. R., and Blake, N. J.: Aerosols from biomass burning over the tropical South Atlantic region: Distributions and impacts, Journal of Geophysical Research: Atmospheres, 101, 24117–24137, doi:10.1029/96JD00717, 1996.
- Chatfield, R. B., Vastano, J. A., Li, L., Sachse, G. W., and Connors, V. S.: The Great African Plume from biomass burning: Generalizations from a three-dimensional study of TRACE A carbon monoxide, Journal of Geophysical Research, 103, 28059–28077, doi:10.1029/97JD03363, 1998.

- Lerot, C., Stavrakou, T., De Smedt, I., Müller, J. F., and Van Roozendael, M.: Glyoxal vertical columns from GOME-2 backscattered light measurements and comparisons with a global model, Atmospheric Chemistry and Physics, 10, 12059–12072, doi:10.5194/acp-10-12059-2010, 2010.
- Meyer-Arnek, J., Ladstätter-Weißenmayer, A., Richter, A., Wittrock, F., and Burrows, J. P.: A study of the trace gas columns of O3, NO2 and HCHO over Africa in September 1997, Faraday Discuss., 130, 387–405, doi:10.1039/b502106p, 2005.
- Ridley, D. A., Heald, C. L., and Ford, B.: North African dust export and deposition: A satellite and model perspective, Journal of Geophysical Research Atmospheres, 117, D022002, doi:10.1029/2011JD016794, 2012.
- Stavrakou, T., Müller, J.-F., De Smedt, I., Van Roozendael, M., Kanakidou, M., Vrekoussis, M., Wittrock, F., Richter, A., and Burrows, J. P.: The continental source of glyoxal estimated by the synergistic use of spaceborne measurements and inverse modelling, Atmospheric Chemistry and Physics, 9, 8431–8446, doi:10.5194/acp-9-8431-2009, 2009.
- Volkamer, R., Baidar, S., Campos, T. L., Coburn, S., DiGangi, J. P., Dix, B., Eloranta, E. W., Koenig, T. K., Morley, B., Ortega, I., Pierce, B. R., Reeves, M., Sinreich, R., Wang, S., Zondlo, M. A., and Romashkin, P. A.: Aircraft measurements of BrO, IO, glyoxal, NO2, H2O, O2-O2 and aerosol extinction profiles in the tropics: Comparison with aircraft-/ship-based in situ and lidar measurements, Atmospheric Measurement Techniques, 8, 2121–2148, doi:10.5194/amt-8-2121-2015, 2015.
- Vrekoussis, M., Wittrock, F., Richter, A., and Burrows, J. P.: Temporal and spatial variability of glyoxal as observed from space, Physical Chemistry Chemical Physics, 9, 4485–4504, doi:10.5194/acp-9-4485-2009, 2009.

**Author reply to Referee #2**

Lisa K. Behrens et al.

June 6, 2019

We thank Referee #2 for carefully reading our manuscript and for the helpful comments which will improve the quality of our manuscript. We will reply to the comments point by point.

Furthermore, we noticed a small mistake in the discussion manuscript. For the AMF calculation in the HCHO satellite retrieval, we accidentally used data from a different CTM (TM4-ECPL). However, for consistency, all retrievals in the manuscript should use the same CTM data as a-priori. Therefore, we have replaced the use of TM4-ECPL by the same MOZART-4 data as was also used in the MAX-DOAS and CHOCHO satellite retrievals. This introduces only small changes compared to the dataset shown in the discussion manuscript. In the revised manuscript, all HCHO and CHOCHO retrievals will use the same MOZART-4 data set.

Legend:

- referee comments
- authors comments
- text in manuscript
- changed text in manuscript

This manuscript describes the detection of HCHO and CHOCHO in African outflow during the COPMAR project in 2016. The authors find elevated HCHO and CHOCHO at higher altitudes, and suggest biomass burning and transport of long-lived precursor VOCs as the source. Overall, the manuscript presents an important set of observations and sufficient preliminary analysis and is suitable for ACP. The following comments should be addressed before publication:

Thank you very much for the positive comments.

**Comments:**

1.) Section 4.5 discusses MOZART outputs which show elevated HCHO and CHOCHO between 3000m and 6000m, while MAX-DOAS shows only elevated HCHO. FLEXPART is then used to investigate potential sources. Why not turn on/off biomass burning in MOZART and see the impact on modeled profiles, or look at MOZART outputs of other tracers?

Turning on/off biomass burning or other traces would be a good approach to investigate the differences between MOZART-4 and MAX-DOAS measurements. However, the MOZART-4 product which we use for the comparison and AMF calculation is an official product provided by NCAR through https://www.acom.ucar.edu/wrf-chem/mozart.shtml. Consequently, we did not run MOZART-4 by ourself and cannot run the model with different initial conditions. Furthermore, we suggested in the manuscript that the gases are trapped by aerosols and than transported. This mechanism cannot be shown by changes in the biomass burning initial conditions of MOZART-4. In our study, we used FLEXPART simulations to gain further knowledge of the origin of the airmass which have been observed. FLEXPART can simulate the transport of such gases trapped by aerosols.

**2.) If MOZART underestimates HCHO south of the equator, how does using incorrect MOZART profiles to calculate MAX-DOAS VCDs influence the retrieval in those regions?**